# JUST A SIMPLE TRANSFORMATION IS ENOUGH FOR DATA PROTECTION IN SPLIT LEARNING

## ABSTRACT

Split Learning (SL) aims to enable collaborative training of deep learning models while maintaining privacy protection. However, the SL procedure still has components that are vulnerable to attacks by malicious parties. In our work, we consider feature reconstruction attacks — a common risk targeting input data compromise. We theoretically claim that feature reconstruction attacks cannot succeed without knowledge of the prior distribution on data. Consequently, we demonstrate that even simple model architecture transformations can significantly impact the protection of input data during SL. Confirming these findings with experimental results, we show that MLP-based models are resistant to state-of-the-art feature reconstruction attacks.

## 1 INTRODUCTION

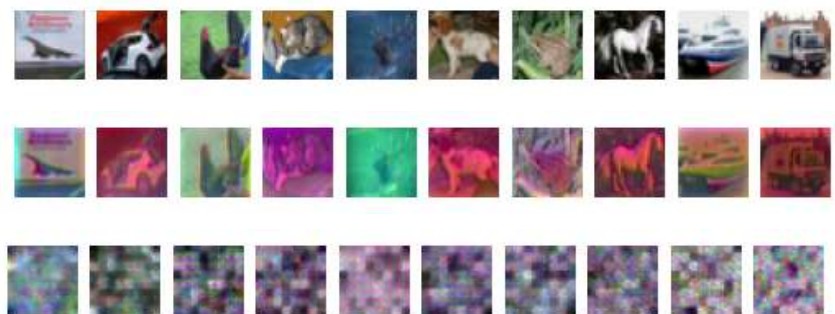

Figure 1: **State-of-the-art feature reconstruction attacks perform remarkably well on CNNs but fail on MLP-based models.** Those are results of the UnSplit attack on CIFAR-10. (**Top**): Original images. (**Middle**): CNN-based client model. (**Bottom**): MLP-Mixer client model.

Federated Learning (FL) (Kairouz et al., 2021; McMahan et al., 2023) introduces a revolutionary paradigm for collaborative machine learning, in which multiple clients participate in cross-device model training on decentralized private data. The key idea is to train the global model without sharing the raw data among the participants. Generally, FL can be divided into two types (Yang et al., 2019a): horizontal (HFL) (Konečný et al., 2017; McMahan et al., 2023), when data is partitioned among clients by samples, and vertical (VFL) (Khan et al., 2023; Liu et al., 2024b; Wei et al., 2022; Yang et al., 2023) when the features of the data samples are distributed among clients. Both HFL and VFL train a global model without sharing the raw data among participants. Since clients in HFL hold the same feature space, the global model is also the same for each participant. Consequently, the FL orchestrator (often termed the server) can receive the parameter updates from each client. In contrast, VFL implies that different models are used for clients, since their feature spaces differ. In this way, the participants communicate through intermediate outputs, called activations.

The focus of this paper is on the privacy concepts of Vertical Federated Learning (Rodríguez-Barroso et al., 2023; Yu et al., 2024; Liu et al., 2024b), namely in Two Party Split Learning (simply, SL) (Gupta & Raskar, 2018; Thapa et al., 2022), where the parties split the model in such a way that the first several layers belong to the client, and the rest are processed at the master server. In SL the client shares its last layer (called Cut Layer) activations, instead of the raw data. As a canonical use case (Sun et al., 2022) of SL, one can think of advertising platform A and advertiser company B. Both parties own different features for each visitor: party A can record the viewing history, while

party B has the visitor's conversion rate. Since each participant has its own private information and they do not exchange it directly, the process of training a recommender system with data from A and B can be considered as Split Learning. In our work, we consider a setting where server holds only the labels, while data is stored on the client side. We discuss in depth the SL setting in § 3.

With regard to practice, the types of attacks from an adversary party are divided into: label inference (Li et al., 2022b; Sun et al., 2022; Liu et al., 2024a; Erdoğan et al., 2022; Kariyappa & Qureshi, 2022), feature reconstruction (Luo et al., 2021; Qiu et al., 2024a; Jin et al., 2022; Geiping et al., 2020; Gupta et al., 2022; Ye et al., 2022; Hu et al., 2022) and model reconstruction (Li et al., 2023; Gao & Zhang, 2023; Fredrikson et al., 2015; Shokri et al., 2017; Driouich et al., 2023; Ganju et al., 2018; Erdoğan et al., 2022). In particular, among all feature reconstruction attacks in Split Learning, we are interested in Model Inversion attacks (MI) (Erdoğan et al., 2022; Fredrikson et al., 2015; He et al., 2019; 2021; Nguyen et al., 2023a;b): one that aims to infer and reconstruct private data by abusing access to the model architecture; and Feature-space Hijacking Attack (for simplicity, we call this type of attacks as "Hijacking" and the attack

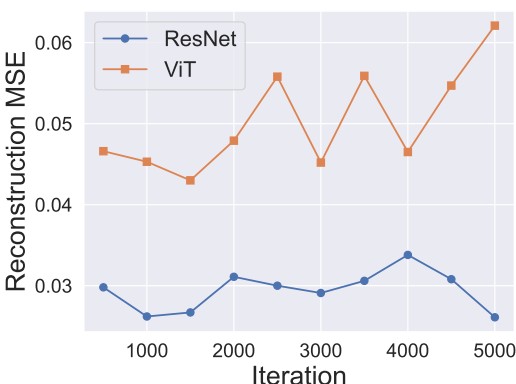

Figure 2: **The best hijacking attack fails to reconstruct features when the client-side model has MLP layers.** This figure demonstrates the dynamics of GAN's reconstruction loss on the Tiny ImageNet dataset (Wu et al., 2017).

from this work we will also call FSHA) (Pasquini et al., 2021; Fu et al., 2023; Yao et al., 2024; Xu et al., 2024): when the malicious party with labels holds an auxiliary dataset from the same domain of the training data of the defending parties; thus, the adversary has prior knowledge of the data distribution.

After revisiting *all the attacks*, to the best of our knowledge, we highlight that state-of-the-art (SOTA) MI and Hijacking attacks (Erdoğan et al., 2022; Pasquini et al., 2021) acquire a knowledge of prior on data distribution (§ 2). Furthermore, these attacks are validated *only on CNN-based* models, bypassing MLP-based models, which also show promise in the same domains. This leads to further questions:

> *(I) Is it that simple to attack features, or does the data prior knowledge give a lot?*
> *(II) Does architectural design play a crucial role in the effectiveness of the latter attacks?*
> *(III) Can we develop a theoretical intuition that MLP models might be more privacy-preserving?*

In this work, we answer these question affirmatively. Following our theoretical justification from § 3.2, by experimentally validating the proposed Hypothesis 1, we reveal that MI (Erdoğan et al., 2022) and Hijacking (Pasquini et al., 2021) attacks fail on MLP-based client-side model. Thus, we neither consider a specific defense framework nor propose a novel method. In contrast, *we demonstrate the failure of feature reconstruction attacks when the architecture is MLP-based*. We summarize our contributions as follows:

(**Contribution 1**) We prove that without additional information about the prior distribution on the data, the feature reconstruction attack in Split Learning cannot be performed even on a one-layer (dense) client-side model. For MLP-based models we state the server's inability to reconstruct the activations in the hidden-space. Furthermore, we provably guarantee that (semi)orthogonal transformations in the client data and weights initialization do not change the transmitted activations during training under the `GD`-like algorithms, and also do not affect convergence for `Adam`-like algorithms.

(**Contribution 2**) We show that Hijacking and Model Inversion attacks fail on MLP-based models *without any additional changes*. We show the effectiveness of our approach against UnSplit (Erdoğan et al., 2022) and Feature-space Hijacking attacks (Pasquini et al., 2021) on popular community datasets (Krizhevsky, 2009; Lecun et al., 1998; Xiao et al., 2017) and argue that feature reconstruction attacks can be prevented without resorting to any of the defenses, while preserving the model

accuracy on the main task. In addition, our findings *can be combined with any of the existing defense frameworks* as this only requires changing the model architecture to an MLP-based one.

(**Contribution 3**) We reconsider the perception of defense quality from a human-side perspective and evaluate resistance against an attacker using the Fréchet inception distance (FID) (Heusel et al., 2017) between the true data and the reconstructed ones. And report the comparison with commonly used MSE in § 4 and § 5.

The code is available at https://anonymous.4open.science/r/JAST-5F3E.

## 2 BACKGROUND & RELATED WORK

Recent feature reconstruction attacks show promising results. Meanwhile, these attacks sometimes require strong assumptions about the capabilities of the attacking side. For example, methods from (Qiu et al., 2024a; Jin et al., 2022; Chen et al., 2024) assume access not only to the architecture, but also to the client-side model parameters during each step of the optimization process (White-Box). The above assumptions rarely occur in real-world applications, as such knowledge is not naturally aligned with the SL paradigm. Nevertheless, an adaptive obfuscation framework from Gu et al. (2023) successfully mitigates the attack from Jin et al. (2022). Moreover, the attacker's setup from these works is more valid for the HFL case (see (Geiping et al., 2020)), where the model is shared among clients and can be trained with (McMahan et al., 2023; Li et al., 2020) algorithms, rather than for VFL. Therefore, such a strong settings are not considered in our work.

**Model Inversion attacks** (Fredrikson et al., 2015; He et al., 2021; 2019; Zhao et al., 2020; Zhu et al., 2019; Wu et al., 2016) are a common threat in machine learning, where an adversary party (server in our case) trains a clone of the client-side model to reconstruct raw data given the client activations. Recent works (Erdoğan et al., 2022; Li et al., 2022a; Fredrikson et al., 2015) demonstrate that Split Learning is also vulnerable to MI attacks. Meanwhile, the most popular defense frameworks (Li et al., 2022a; Sun et al., 2021), aiming to protect data from MI attack, are effective against the adversary with White-Box access, which does not hold in real-world, and require imitation of the attacker (called attacker-aware training) using client-side inversion models, which leads to a 27% floating point operations (FLOPs) computational overhead(see Li et al. (2022a) Table 6).

Next, we come to Unsplit, proposed in Erdoğan et al. (2022), the main MI attack aiming to reconstruct input image data by exploiting an extended variant of coordinate descent (Wright, 2015). Given the client model $f$, its clone $\tilde{f}$ (i.e., the randomly initialized model with the same architecture), the adversary server attempts to solve the two-step optimization problem:

$$\tilde{X}^* = \arg\min_{\tilde{X}} \mathcal{L}_{\text{MSE}}\left(\tilde{f}(\tilde{X}, \tilde{W}), \; f(X, W)\right) + \lambda \text{TV}(\tilde{X}), \tag{1}$$

$$\tilde{W}^* = \arg\min_{\tilde{W}} \mathcal{L}_{\text{MSE}}\left(\tilde{f}(\tilde{X}, \tilde{W}), \; f(X, W)\right). \tag{2}$$

In this context, $X, W$ represent the client model's private inputs and parameters; TV denotes the total variation distance (Rudin et al., 1992) for image pixels (this term allows the attacker to use prior on data distribution); and $\tilde{X}^*, \tilde{W}^*$ are the desired variables for the attacker's reconstructed output and parameters, respectively. Whereas, $\lambda$ is the coefficient to modify the impact of the total variation, e.g., minimizing $\text{TV}(\tilde{X})$ results in smoother images. At the beginning of the attack, "mock" features $\tilde{X}$ initializes as a constant matrix. It should be noted that this optimization process can be applied both before and after training $f$. The latter corresponds to feature reconstruction during the inference stage. The authors assume that the server is only aware of the architecture of the client model $f$.

**The Feature-space Hijacking attack (FSHA)** was initially proposed in Pasquini et al. (2021). The authors mention that the server's ability to control the learning process is the most pervasive vulnerability of SL, which is not used in UnSplit setting. Indeed, since the server is able to guide the client model $f$ towards the required functional states, it has the capacity to reconstruct the private features $X$. In hijacking attacks (Fu et al., 2023; Pasquini et al., 2021; Yu et al., 2023), the malicious server exploits an access to a public dataset $X_{\text{pub}}$ of the same domain as $X$ to subdue the training protocol.

Specifically, in (Pasquini et al., 2021), the server initializes three additional models: encoder $\psi_{\text{E}}$, decoder $\psi_{\text{D}}$ and discriminator $D$. While the client-side model $f : \mathcal{X} \to \mathcal{Z}$ is initialized as a

mapping between the data distribution $\mathcal{X}$ and a hidden-space $\mathcal{Z}$, the encoder network $\psi_{\mathrm{E}} : \mathcal{X} \to \tilde{\mathcal{Z}}$ dynamically defines a function to certain subset $\tilde{\mathcal{Z}} \subset \mathcal{Z}$. Since the goal is to recover $X \in \mathcal{X}$, to ensure the invertibility of $\psi_{\mathrm{E}}$, the server trains the decoder model $\psi_{\mathrm{D}} : \mathcal{Z} \to \mathcal{X}$. To guide $f$ towards learning $\tilde{\mathcal{Z}}$, server uses a discriminator network trained to assign high probability to the $\psi_{\mathrm{E}}(X_{\mathrm{pub}})$ and low to the $f(X)$.

The general scheme of the attack is the following:

$$\psi_{\mathrm{E}}^*, \ \psi_{\mathrm{D}}^* = \arg \min_{\psi_{\mathrm{E}}, \psi_{\mathrm{D}}} \mathcal{L}_{\mathrm{MSE}} \left( \psi_{\mathrm{D}}(\psi_{\mathrm{E}}(X_{\mathrm{pub}})), \ X_{\mathrm{pub}} \right), \tag{3}$$

$$D = \arg \min_{\mathrm{D}} \left[ \log(1 - D(\psi_{\mathrm{E}}(X_{\mathrm{pub}}))) + \log(D(f(X))) \right],$$

$$\mathcal{L}^* = \arg \min_{f} \left[ \log \left( 1 - D(f(X)) \right) \right].$$

And, finally, server recovers features with:

$$\tilde{X}^* = \psi_{\mathrm{D}}^* \left( \mathcal{L}^*(X) \right). \tag{4}$$

This paper has led to the creation of other works that study FSHA. Erdogan et al. (2022) propose a defense method SplitGuard in which the client sends fake batches with mixed labels with a certain probability. Then, the client analyzes the gradients corresponding to the real and fake labels and computes SplitGuard score to assess whether the server is conducting a Hijacking Attack and potentially halt the training. In response to the SplitGuard defense, Fu et al. (2023) proposed SplitSpy: where it is observed that samples from the batch with the lowest prediction score are likely to correspond to the fake labels and should be removed during this round of FSHA. Therefore, SplitSpy computes gradients from discriminator $D$ only for survived samples. We would like to outline that this attack uniformly weaker compared to the original FSHA (Pasquini et al., 2021) in the absence of the SplitGuard defense. Thus, we will only consider this attack later.

**Quality of the defense.** Sun et al. (2023) studies the faithfulness of different privacy leakage metrics to human perception. Crowdsourcing revealed that hand-crafted metrics (Sara et al., 2019; Pedersen & Hardeberg, 2012; Zhang et al., 2018; Wang & Bovik, 2002) have a weak correlation and contradict with human awareness and similar methods (Zhang et al., 2018; Huynh-Thu & Ghanbari, 2008). From this point of view, we reconsider the usage of the MSE metric for the evaluation of the defense against feature reconstruction attacks, i.e., the quality of reconstruction. Given that the main datasets contain images, we suggest to rely on Frechet Inception Distance (FID) (Heusel et al., 2017). Besides the fact that MSE metric is implied into the attacker algorithms (Equations 1, 2, 3), most of works on evaluation of the images quality rely on FID. From the privacy perspective, the goal of the successful defense evaluation is to compare privacy risks of a classification model under the reconstruction attack. This process can be formalized for Split Learning in the following way: let the attack mechanism $\mathcal{M}$ aiming to reconstruct client model $f$ data $X$ given the Cut Layer outputs $H$, depending on the setup, $\mathcal{M}$ can access the client model architecture (in other settings this assumption may differ), then the privacy leakage is represented as

$$\mathrm{PrivacyLeak} = \mathrm{InfoLeak}\left( X, \mathcal{M}(H, f) \right) = \mathrm{PrivacyLeak}\left( X_{\mathrm{rec}} \right),$$

where InfoLeak stands for the amount of information leakage in reconstructed images $X_{\mathrm{rec}}$. Note that, $\mathcal{M}$ receives the Cut Layer outputs $H$ at every iteration; then, the PrivacyLeak can also be measured during every iteration of the attack. Generally, information leakage can be represented through the hand-crafted metric $\rho$: $\mathrm{InfoLeak} = \rho(X, X_{\mathrm{rec}})$.

## 3 PROBLEM STATEMENT AND THEORETICAL MOTIVATION

In this section, we: (I) outline the (Two Party) Split Learning setting; (II) demonstrate that (semi)orthogonally transformed data and weights result in an identical training process from the server's perspective (**Lemma 1**); (III) prove that in this scenario, even a malicious server cannot reconstruct features without prior knowledge of the data distribution (**Lemma 2**); (IV) show that similar reasoning applies to the distribution of activations before the Cut Layer (**Lemma 3**); (V) propose **Hypothesis 1** explaining why SOTA feature reconstruction attacks achieve significant success and suggest potential remedies.

**Notation.** We denote the client's model in SL as $f$, with the weights $W \in \mathbb{R}^{d \times d_{\mathrm{h}}}$. Under $X$, we consider a design matrix of shape $\mathbb{R}^{n \times d}$. We denote activations that client transmits to the server

as $H \in \mathbb{R}^{\mathrm{n} \times \mathrm{d_h}}$, while $\mathcal{Z}$ and $\mathcal{X}$ are the hidden-space and the data distribution, respectively. $\mathrm{n}$ corresponds to the number of samples in dataset $X$, $\mathrm{d}$ stands for the features belonging to client and $\mathrm{d_h}$ is a hidden-size of the model. $\mathcal{L}$ denotes the *loss function of the entire model* (both server and client). Next, we provide a detailed description of our setup.

**Setup.** From the perspective of one client, it cannot rely on any information about the other parties during VFL. Then, to simplify the analysis, we consider the Two Party Split Learning process. Server $\mathrm{s}$ (label-party) holds a vector of labels $y$, while the other data is located at the client-side matrix $X$. Server and client have their own neural networks. In each iteration, the non-label party computes activations $H = f(X, W)$ and sends it to the server. Then, the remaining forward computation is performed only by server $f_s$ and results in the predictions $p = f_s(H)$ and, consequently, to the loss $\mathcal{L}(p, y)$. In the backward phase, client receives $\frac{\partial \mathcal{L}}{\partial H}$, and computes the $\frac{\partial \mathcal{L}}{\partial W} = \frac{\partial \mathcal{L}}{\partial H} \frac{\partial H}{\partial W}$.

### 3.1 MOTIVATION I: ORTHOGONAL TRANSFORMATION OF DATA & WEIGHTS STOPS ATTACK

In this part, we consider client $f$ as *one-layer* linear model $f = XW$ with $W \in \mathbb{R}^{\mathrm{d} \times \mathrm{d_h}}$. Note that *(semi)orthogonal* transformations $X \to XU$, $W_0 \to U^\top W_0$ preserve the outputs of $f$ at initialization. Turns out, that it also holds for subsequent iterations of (Stochastic) Gradient Descent:

> **Lemma 1.** *For a one-layer linear model trained using* `GD` *or* `SGD`*, there exist continually many pairs of client data and weights initialization that produce the same activations at each step.*

The complete proof of this lemma is presented in Appendix A.1. These pairs have the form $\{\tilde{X}, \tilde{W}_0\} = \{XU, U^\top W_0\}$, where $U$ — arbitrary orthogonal matrix. The use of such a pair within `(S)GD` induces the same rotation of the iterates: $\tilde{W}_+^{\mathrm{GD}} = \tilde{W} - \gamma \partial \mathcal{L} / \partial \tilde{W} = U^\top (W - \gamma X^\top \partial \mathcal{L} / \partial H) = U^\top W_+^{\mathrm{GD}}$. With such orthogonal transformation, the client produces *the same activations*, as if we had left $X$ and $W$ unchanged: $\tilde{X}\tilde{W} = H = XW$. The server cannot distinguish between the different data distributions that produce identical outputs; therefore, the true data also cannot be obtained. This results in:

> **Remark 1.** *Under the conditions of Lemma 1, if the server has no prior information about the distribution of $X$, the label party cannot reconstruct initial data $X$ (only up to an arbitrary orthogonal transformation).*

Recent work (Ye et al., 2022) states similar considerations, but their remark about `Adam` (Kingma & Ba, 2017) and `RMSprop` (Graves, 2014) not changing the Split Learning protocol is false. The use of `Adam` or `RMSprop` with (semi)orthogonal transformations changes the activations produced by the SL protocol because of the "root-mean-square" dependence of the denominator on gradients in their update rules. We delve into this question in Remark 4 (see Appendix A for details). In fact, Lemma 1 holds only for algorithms whose update step is *linear* with respect to the gradient history.

However, while `Adam` and `RMSProp` do not preserve the SL protocol in terms of exact matching of transmitted activations, we can relax the conditions and consider the properties of these algorithms from the perspective of "protocol preservation" in the sense of maintaining convergence to the same value. To begin with, let us note the following:

> **Remark 2.** *The model's optimal value $\mathcal{L}^*$ after Split Learning is the same for any orthogonal data transformation. Indeed, $\forall \tilde{X} = XU \ \exists \tilde{W}^* = U^\top W^* : \ \mathcal{L}(\tilde{X}, \tilde{W}^*) = \mathcal{L}^* = \mathcal{L}(X, W^*)$. Thus, $\mathcal{L}^*$ remains the same if we correspondingly rotate the optimal weights.*

In the case of a convex or strongly-convex function (entire model) $\mathcal{L}(X, W)$, the optimal value $\mathcal{L}^*$ is unique, and therefore any algorithm is guaranteed to converge to $\mathcal{L}^*$ for any data transformation. Meanwhile, for general non-convex functions, convergence behavior becomes more nuanced: in fact, in Example 1, we present a function on which the `Adam` algorithm converges before data and weight transformation and diverges after the transformation. However, the situation changes when we turn to functions satisfying the Polyak-Łojasiewicz-condition (PL) [1], which is used as a canonical description of neural networks (Liu et al., 2021). We, then, provably claim that the SL protocol is preserved for PL functions with orthogonal transformations of data and weights. We show that

---

[1] Note that PL-condition does not imply convexity, see footnote 1 from (Li et al., 2021).

`Adam`'s preconditioning matrices can be bounded regardless of the $W$ initialization, and derive a Descent Lemma 5 with the modification of bounded gradient Assumption 3 similar to prior works (Sadiev et al., 2024; Défossez et al., 2022). The convergence guarantees are covered in Lemma 6 and the theoretical evidence can be found in Appendix A.4.

According to Lemma 1, the desired result of the defending party can be achieved if we weaken the attacker's knowledge of prior on data distribution (see § 3.2). Actually, even knowledge of weights does not help the attacker:

> **Corollary 1.** *Under the conditions of Lemma 1, assume that server knows the first layer $W_1$ of $f$, and let this layer be an invertible matrix. Then, the label party cannot reconstruct the initial data $X$ (only up to an arbitrary orthogonal transformation).*

Indeed, the activations send to the server in the first step: $H_1 = XW_1$, but if the client performs an orthogonal transformation leading to $\tilde{X}$, then, server can recover only $\tilde{H}_1 W_1^{-1}$, where $\tilde{H}_1 = \tilde{X} W_1$. Meanwhile, the difference between $X$ and $\tilde{X}$ affects only the initialization of weights, and thus should not change the final model performance much. Next, we conclude that *even a malicious server* cannot reconstruct the client's data without the additional prior on $X$.

> **Lemma 2.** *Under the conditions of Lemma 1, assume training with the malicious server sending arbitrary vectors instead of real gradients $G = \partial\mathcal{L}/\partial H$. In addition, the server knows the initialization of the weight matrix $W_1$. Then, if the client applies a non-trainable orthogonal matrix before $W_1$, the malicious server cannot reconstruct initial data $X$ (only up to an arbitrary orthogonal transformation).*

> **Remark 3.** *With the same reasons as for Lemma 1, if even the malicious server from Lemma 2 has no prior information about the distribution of $X$, it is impossible for the label party to reconstruct the initial data $X$.*

We note that Lemmas 1 and 2 are correct under their conditions even if the update of the client-side model includes bias, see Corollary 2.

## 3.2 MOTIVATION II: YOU CANNOT ATTACK THE ACTIVATIONS BEFORE "CUT LAYER"

Up until now, we considered the client-side model with one linear layer $W$ and proved that orthogonal transformation of data $X$ and weights $W$ lead to the same training protocol. The intuition behind Lemmas 1 and 2 suggests that in the client model, one should look for layers whose inputs cannot be given the prior distribution. Meanwhile, the activations after each layer are also data on which the server's model trains; however, this data become "smashed" after being processed by $f$. This brings us to the consideration of Cut Layer, since this is a "bridge" between the client and server. The closer Cut Layer is to the first layer of the client's model $f$, the easier it is to steal data (Erdoğan et al., 2022; Li et al., 2022a); the complexity of attack increase with the "distance" between Cut Layer and data. Consequently, we pose the following question: "Does our intuition from Remark 1 apply to the activations before Cut Layer?"

> **Lemma 3.** *[Cut Layer Lemma] There exist continually many distributions of the activations before the linear Cut Layer that produce the same Split Learning protocol.*

The results of Lemma 3 lead to a promising remark. While server might have prior on original data distribution, acquiring a prior on the distribution of the activations before Cut Layer is, generally, much more challenging. The absence of knowledge regarding the prior distribution of activations, combined with the assertion in Lemma 3, yields a result for activations analogous to Remark 1. Specifically, even with knowledge of a certain prior on the data, the server can, at best, reconstruct *activations* only up to an orthogonal transformation [2].

## 3.3 SHOULD WE USE DENSE LAYERS AGAINST FEATURE RECONSTRUCTION ATTACKS?

The findings from § 3.2 indicate that the reconstruction of activations poses significant challenges for the server. However, many feature reconstruction attacks achieve considerable success. This raises

---

[2]Excluding degenerate cases, such as when the server knows that the client's network performs an identity transformation.

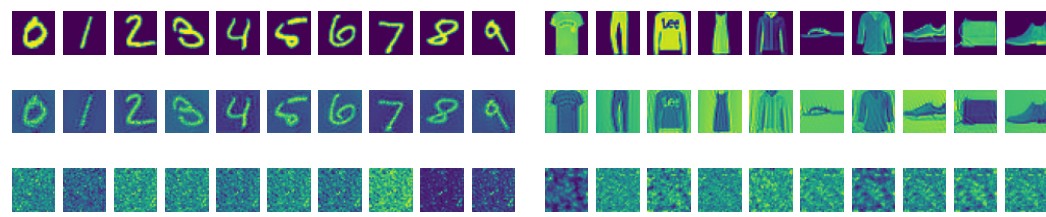

Figure 3: Results of UnSplit attack on MNIST. Figure 4: Results of UnSplit attack on F-MNIST. (**Top**): Original images. (**Middle**): CNN-based (**Top**): Original images. (**Middle**): CNN-based client model. (**Bottom**): MLP-based client model. client model. (**Bottom**): MLP-based client model.

the question: "Does the server's inability to reconstruct activations before the Cut Layer not impede its capacity to reconstruct data features?" Alternatively, "Could it be that the conditions of Lemma 3 do not hold in practical scenarios?"

To investigate this matter more thoroughly, we examined outlined in § 2 feature reconstruction attacks. Specifically, we focused on UnSplit (Erdoğan et al., 2022) and FSHA (Pasquini et al., 2021), which are SOTA representatives (to the best of our knowledge) of the Model Inversion and Feature Space Hijacking attack categories, respectively. UnSplit requires knowledge of the client-side model architecture, while FSHA should know the dataset $X_{\text{pub}}$ of the same distribution as the original $X$. Assumptions are quite strong in the general case, but, we, in turn, argue that their attacks can be mitigated *without any additional modifications* in UnSplit (Erdoğan et al., 2022) and FSHA (Pasquini et al., 2021) assumptions (see § 4). Both of these attacks are validated exclusively on image datasets, utilizing CNN architectures. Consequently, the client-side model architectures *lack fully connected (dense) layers before Cut Layer and the conditions of Lemma 3 do not hold.*

While a convolutional layer is inherently a linear operation and can be represented as matrix multiplication — where the inputs and weights can be flattened into 2D tensors — the resulting matrix typically has a very specific structure. In particular, all elements except for $(kernel\ size) \cdot (kernel\ size)$ entries in each row are zero. Therefore, an inverse transform does not exist in a general sense — meaning not every matrix multiplication can be expressed as a convolution, as the resultant matrix generally contains significantly more non-zero elements. As a result, "merging" an orthogonal matrix into a sequence convolutional layers and poolings by multiplying the convolution weights with an orthogonal matrix is impossible, since this would result in a matrix with an excess of non-zero elements. Based on this observation we propose:

> **Hypothesis 1.** *Could it be that the attacks are successful due to the lack of dense layers in the client architecture? Will usage of MLP-based architectures for $f$, instead of CNNs, be more privacy preserving against Model Inversion attack and FSHA?*

We intend to experimentally test this conjecture in the following section.

## 4 EXPERIMENTS

This section is dedicated to the experimental validation of the concepts introduced earlier. To test our Hypothesis 1, we evaluate the effectiveness of UnSplit and FSHA on popular datasets (Krizhevsky, 2009; Lecun et al., 1998; Xiao et al., 2017) in setting where at least one dense layer on the client side. It is important to note that although MLP-based architecture may not be conventional in the field of Computer Vision (where CNN usage is more prevalent), dense layers are the backbone of popular model architectures in many other Deep Learning domains, such as Natural Language Processing, Reinforcement Learning, Tabular Deep Learning, etc. In these domains, dense layers are commonly found at the very start of the architecture, and thus, when the network is split for VFL training, these layers would be contained in $f$. Furthermore, even within the Computer Vision field, there is a growing popularity of architectures like Vision Transformers (ViT) (Dosovitskiy et al., 2021) and MLP-Mixer (Tolstikhin et al., 2021), which also incorporate dense layers at the early stages of data processing. Therefore, we contend that with careful architectural selection, integrating dense layers on the client side should not lead to a significant deterioration in the model's utility score.

**UnSplit.** Before delving into the primary experiments of our study, we must note that unfortunately we were unable to fully reproduce the results of UnSplit using the code from their repository. Specifically, the images reconstructed through the attack were significantly degraded when deeper Cut Layers

were used (see column "Without Noise" in Table 4). However, for the case where $cut\ layer = 1$ (i.e., when there is only one layer on the client side), the images were reconstructed quite well. Therefore, we used this setup for our comparisons. As previously mentioned, to test Hypothesis 1, we utilized an MLP model with single or multiple dense layers on the client side (in the experiments below, client's part holds *only* one-layer model). For CIFAR-10, we use MLP-Mixer, which maintains the performance of a CNN-based model while incorporating dense layers into the design. The results of the attack are shown in Figures 3 and 4. Despite our efforts to significantly increase the $\lambda$ parameter in the TV in Equation (1) up to $100$ — thereby incorporating a stronger prior about the data into the attacker's model — the attack failed to recover the images, thus supporting the assertion of Lemma 1.

| Dataset | Model | MSE $\mathcal{X}$ | MSE $\mathcal{Z}$ | FID | Acc% |
|---------|-------|-------|-------|-----|------|
| MNIST | MLP-based | 0.27 | $3 \cdot 10^{-8}$ | 394 | 98.42 |
| | CNN-based | 0.05 | $2 \cdot 10^{-2}$ | 261 | 98.68 |
| F-MNIST | MLP-based | 0.19 | $4 \cdot 10^{-5}$ | 361 | 88.31 |
| | CNN-based | 0.37 | $4 \cdot 10^{-2}$ | 169 | 89.23 |
| CIFAR-10 | MLP-Mixer | 1.398 | $6 \cdot 10^{-6}$ | 423 | 89.29 |
| | CNN-based | 0.056 | $4 \cdot 10^{-3}$ | 455 | 93.61 |

Table 1: **UnSplit attack on MNIST, F-MNIST, and CIFAR-10 datasets.**

Additionally, Table 1 presents the reconstruction loss values between normalized images. Here MSE $\mathcal{X}$ and FID shows the difference between the original and reconstructed images, and MSE $\mathcal{Z}$ refers to the loss between the activations $H = f(X, W)$ and $\tilde{H} = \tilde{f}(\tilde{X}, \tilde{W})$. Acc% denotes final accuracy of the trained models. As we can see, the results of MLP-based model are very close to its CNN-based counterpart. In the image space $\mathcal{X}$, FID appears to be a superior metric compared to MSE for accurately capturing the consequences of the attack. Furthermore, the tables show the MSE between activations before the Cut Layer for both the original and reconstructed images. These results indicate that in the case of the dense layer, the activations almost completely match, with significantly lower MSE than those even for well-reconstructed images. This implies that while the attack can perfectly fit $H = XW$ (in notation, when $cut\ layer = 1$), it fails to accurately recover $X$. In addition, we note that even a one-layer linear model remains resistant to the UnSplit attack, and we provide the corresponding experiments in Appendix C.

**FSHA.** Similarly to the previous subsections, we replaced the client's model in the FSHA attack(Pasquini et al., 2021) with an MLP consisting of one or multiple layers. The attacker's models also varied, ranging from ResNet(He et al., 2015) architectures (following the original paper) to MLPs, ensuring that the attacker's capabilities are not constrained by the limitations of any architectural design. The results, illustrated in Figures 5 and 6, consistently demonstrate that the malicious party fully reconstructs the original data in the case of the ResNet architecture and completely fails in the case of the Dense layer. In addition to the reconstructed data shown in Figure 7, we computed the Reconstruction error (Equation (4)) and Encoder-Decoder error (Equation (3)) for a client using a ResBlock architecture (as in the original paper) and a client employing an MLP-based architecture. These plots reveal that the Encoder-Decoder pair for both architectures is equally effective at reconstructing data from the public dataset on the attacker's side. However, a challenge arises on the attacker's side with the training of GAN(Goodfellow et al., 2014). It is evident that in the presence of a Dense layer on the client side, the GAN fails to properly align the client's model representation within the required subset of the feature space. Instead, it converges to mapping models of all classes into one or several modes within the activation space, corresponding to only a few original classes. This phenomenon is particularly well illustrated for the F-MNIST dataset in Figure 6. To scale up this experiment, we run FSHA on Tiny ImageNet data comparing ResNet-18 and ViT client-side models. We report the results in Figure 2 and Appendix C.

**Evaluation with FID.** Inspired by prior works on GANs(Goodfellow et al., 2014), we apply FID to the InfoLeak scheme for the next reasons: (I) FID measures the information leakage as the distribution difference between between original and reconstruction images, thus InfoLeak$(X, X_{rec}) \propto$ FID$(X, X_{rec})$. (II) Usage of FID is a more common approach when dealing with images. (III) The widespread metric in reconstruction evaluation is MSE, that lacks an interpretation for complex images (Sun et al., 2023), at least from the CIFAR-10(Krizhevsky, 2009) dataset. However, we notice that the privacy evaluation of feature reconstruction attacks requires refined. The values of FID and MSE in Table 1 suggest that FID is a more accurate reflection of the attack's outcomes than

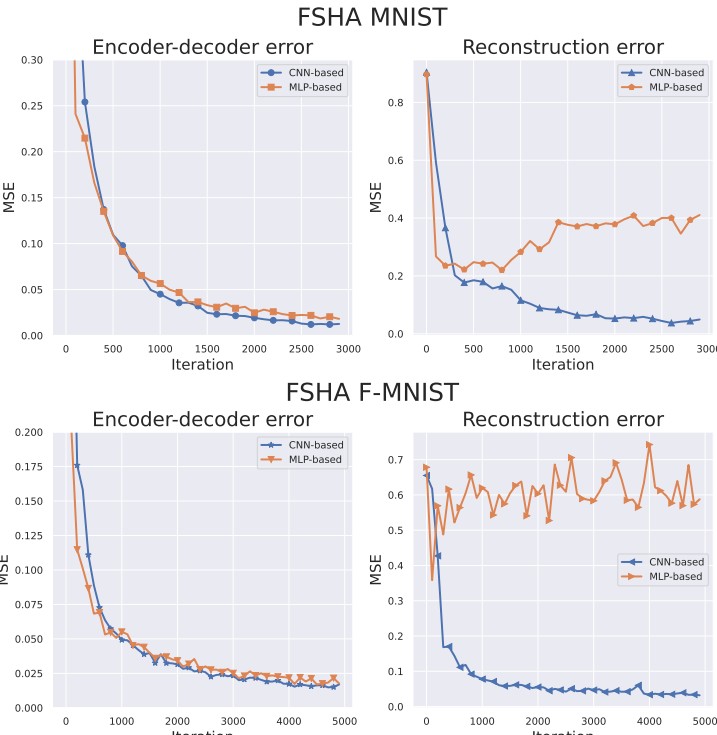

Figure 5: Results of FSHA attack on MNIST. (**Top**): Original images. (**Middle**): CNN-based client model. (**Bottom**): MLP-based client model.

Figure 6: Results of FSHA attack on F-MNIST. (**Top**): Original images. (**Middle**): CNN-based client model. (**Bottom**): MLP-based client model.

Figure 7: **When the dense layer resides on the client side, the GAN in the FHSA attack cannot align the client's representation within the feature space.** This phenomenon is clearly visible in the plots of reconstruction error, which diverge only for MLPs. We point out that replacing the MLP with a smaller model yields the same outcome.

MSE in the image space $\mathcal{X}$. For instance, on the F-MNIST dataset, the MSE is higher for a CNN architecture despite the better quality of the reconstructed images. This discrepancy appears to stem from differences in background pixel values compared to the original images.

## 5 DISCUSSIONS

With our work, we contribute to a better understanding of the meaningfulness of feature reconstruction attacks. We show that the architectural design of client-side model reflects the attack's performance. Particularly, even the most powerful Black-Box feature reconstruction attacks fail when attempting to compromise client's data when its architecture is MLP. We observe our findings experimentally, and provide a rigorous mathematical explanation of this phenomenon. Our study contributes to recent advances in privacy of VFL (SL) and suggest that novel Black-Box attacks should be revisited to address the challenges which occurs with MLP-based models. We note that our approach may not be impactful on NLP tasks, since the language models require a discrete input instead of the continuous which we actively exploit during the theoretical justifications and experiments with MLP-based models. However, we note that UnSplit attack also cannot be efficiently performed against the transformer-based architectures due to the huge amount of computational resources for training multi-head attention and FFN layers with the coordinate descent.

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

# A  APPENDIX

## CONTENTS

## A.1 ORTHOGONAL MATRICES GIVE IDENTICAL TRAINING FOR DIFFERENT DATA DISTRIBUTIONS

**The proof of Lemma 1**.

*Proof.* To proof this claim we will show that it is possible to obtain two completely identical model training procedures from the server's perspective with different pairs of the client data. Moreover, there are continually many such pairs.

Let the client's data come from the distribution $\mathcal{X}$ and the initialization of the $W$ matrix be $W_1$.

Then, consider an orthogonal matrix $U \in \mathbb{R}^{d \times d}$, distribution $\tilde{\mathcal{X}} = \mathcal{X}U$ and initialization $\tilde{W}_1 = U^\top W_1$.

We now show by induction, that in each step $k$ of the model training protocol the client will obtain the same latents $H_k$, $\tilde{H}_k$, both with $\{\mathcal{X}, W_1\}$ and $\{\tilde{\mathcal{X}}, \tilde{W}_1\}$ pairs, respectively, and, therefore, will transfer the same activations to the server.

Note that we make no assumptions about the data $X_k$ processed at each step of optimization. It can be either the entire dataset or a mini-batch of arbitrary size.

1. **Base case,** $k = 1$: $H_1 = X_1 W_1 = X_1 U U^\top W_1 = \tilde{X}_1 \tilde{W}_1 = \tilde{H}_1$

2. **Induction step,** $k + 1 > 1$: Let $H_k = \tilde{H}_k$ by induction hypothesis. Then $\partial \mathcal{L}/\partial H_k = \partial \mathcal{L}/\partial \tilde{H}_k = G_k \in \mathbb{R}^{n \times d_h}$. Recall, that

$$\frac{\partial \mathcal{L}}{\partial W_k} = \frac{\partial \mathcal{L}}{\partial H_k} \frac{\partial H_k}{\partial W_k} = X_k^\top \frac{\partial \mathcal{L}}{\partial H_k} = X_k^\top G_k.$$

Then the step of GD for the pairs $\{\mathcal{X}, W_1\}$ and $\{\tilde{\mathcal{X}}, \tilde{W}_1\}$ returns

$$W_{k+1} = W_k - \gamma X_k^\top G_k$$

and

$$\tilde{W}_{k+1} = \tilde{W}_k - \gamma \tilde{X}_k^\top G_k = U^\top W_k - \gamma U^\top X_k^\top G_k$$

respectively.

Thus, at $k + 1$ step

$$\begin{aligned} H_{k+1} = X_{k+1} W_{k+1} &= X_{k+1} W_k - \gamma X_{k+1} X_k^\top G_k = \\ &= X_{k+1} U U^\top W_k - \gamma X_{k+1} U U^\top X_k^\top G_k = \\ &= \tilde{X}_{k+1} \tilde{W}_k - \gamma \tilde{X}_{k+1} \tilde{X}_k^\top G_k = \\ &= \tilde{X}_{k+1} \tilde{W}_{k+1} = \tilde{H}_{k+1}, \end{aligned}$$

i.e., the activations sent to the server are identical for $\{\mathcal{X}, W_1\}$, $\{\tilde{\mathcal{X}}, \tilde{W}_1\}$ pairs.

Since the above proof is true for any orthogonal matrix $U \in \mathbb{R}^{d \times d}$, there exist continually many different pairs of client data that produces the same model training protocol result.

$\square$

## A.2 FAKE GRADIENTS

**The proof of Lemma 2.**

*Proof.* For simplicity, assume training with a full-batch gradient: $\forall k \ X_k = X$.

It is noteworthy that the proof remains equivalent for gradients computed w.r.t. both batch and full-batch scenarios.

Also let us write $X$ instead of the $X$ and denote an arbitrary vectors sent by server as $G^{\text{fake}}$.

Consequently, client transmits $H_{k+1} = XW_{k+1}$ in each step.

Then, the following is true for the weight matrix:

$$
\begin{aligned}
W_{k+1} = W_k - \gamma X^\top G_k^{\text{fake}} &= \\
&= \left(W_{k-1} - \gamma X^\top G_{k-1}^{\text{fake}}\right) - \gamma X^\top G_k^{\text{fake}} = \\
&= \cdots = W_1 - \gamma X^\top \left[\sum_{i=1}^k G_i^{\text{fake}}\right].
\end{aligned}
$$

$$
H_{k+1} = XW_{k+1} = XW_1 - \gamma XX^\top \left[\sum_{i=1}^k G_i^{\text{fake}}\right],
$$

$$
\tilde{H}_{k+1} = \tilde{X}W_1 - \gamma \tilde{X}\tilde{X}^\top \left[\sum_{i=1}^k G_i^{\text{fake}}\right] = \tilde{X}W_1 - \gamma XX^\top \left[\sum_{i=1}^k G_i^{\text{fake}}\right].
$$

The only term that has changed is the first summand $\tilde{X}W_1$.

Similarly to the previous case, the server can recover $\tilde{X}$. But now it cannot recover the real data $X$.

Indeed, the server can only build its attack based on the knowledge of $\tilde{X} = XU$ and $\tilde{X}\tilde{X}^\top$.

This means that it cannot distinguish between two different pairs $\{\mathcal{X}, U\}$ if they generate the same values $\tilde{X}\tilde{X}^\top$.

Notice that, by simply multiplying a given pair $\{\mathcal{X}, U\}$ by another orthogonal matrix $V$ it is possible to construct a continuum of other pairs $\{\hat{\mathcal{X}} = \mathcal{X}V, \hat{U} = V^\top U\}$ that produce the same values $\tilde{X}$, $\tilde{X}\tilde{X}^\top$:

$$
\{\mathcal{X}, U\} \to \{\hat{\mathcal{X}} = \mathcal{X}V, \hat{U} = V^\top U\}
$$

$$
\tilde{X} = XU = (XV)(V^\top U) = \hat{X}\hat{V},
$$

and

$$
\begin{aligned}
\tilde{X}\tilde{X}^\top &= (XU)(U^\top X^\top) \\
&= (XVV^\top U)(U^\top VV^\top X^\top) \\
&= (\hat{X}\hat{U})(\hat{U}^\top \hat{X}^\top) = \hat{X}\hat{X}^\top,
\end{aligned}
$$

correspondingly.

Therefore, even malicious server that sabotages the protocol cannot distinguish between all the possible $\mathcal{X}U$ distributions.

In addition we should note, that multiplying $X$ by some orthogonal matrix $U$ should not affect the quality of training, because the only term that contains $U$ in the final formula for activations $H$ is the first summand $\tilde{X}W_1 = XUW_1$.

However we can rewrite this formula as $XUW_1 = X\tilde{W}_1$, where $\tilde{W}_1 = UW_1$. In this way, it is as if we have changed the initialization $W_1$ of the weight matrix, that usually do not affect final model quality.

$\square$

Before we delve into the strongest lemmas, we also note, that the results of Lemmas 1 and 2 holds for the one-layer linear client-side model with bias, i.e., when $H_{k+1} = X_{k+1}W_{k+1} + B_{k+1}$.

> **Corollary 2.** *Lemmas 1 and 2 are correct under their conditions even if the update of a client-side model includes bias.*

*Proof.* It is sufficiently to show only for Lemma 2.

For simplicity, assume training with a full-batch gradient: $\forall k \; X_k = X$, since that the proof remains equivalent. Let us denote an arbitrary vectors sent by server as $G^{\text{fake}}$.

Client transmits $H_{k+1} = XW_{k+1} + B_{k+1}$ in each step.

Weights update:

$$
\begin{aligned}
W_{k+1} &= W_k - \gamma X^\top G_k^{\text{fake}} = \\
&= \left(W_{k-1} - \gamma X^\top G_{k-1}^{\text{fake}}\right) - \gamma X^\top G_k^{\text{fake}} = \\
&= \cdots = W_1 - \gamma X^\top \left[\sum_{i=1}^{k} G_i^{\text{fake}}\right].
\end{aligned}
$$

Biases update:

$$
B_{k+1} = B_k - \gamma \frac{\partial \mathcal{L}}{\partial B_k} = B_k - \gamma \frac{\partial \mathcal{L}}{\partial H_k} \frac{\partial H_k}{B_k} = B_k - \gamma I \frac{\partial \mathcal{L}}{\partial H_k} = \cdots = B_1 - \gamma \left[\sum_{i=1}^{k} \frac{\partial \mathcal{L}}{\partial H_i}\right],
$$

recall that the serve transmits the fake gradient $G_i^{\text{fake}}$ instead of $\frac{\partial \mathcal{L}}{\partial H_i}$, thus

$$
B_{k+1} = B_1 - \gamma \left[\sum_{i=1}^{k} G_i^{\text{fake}}\right].
$$

Activations update:

$$
\begin{aligned}
H_{k+1} &= XW_{k+1} + B_{k+1} = XW_1 + B_1 - \gamma \left[I + XX^\top\right] \sum_{i=1}^{k} G_i^{\text{fake}} \\
&= H_1 - \gamma \left[I + XX^\top\right] \sum_{i=1}^{k} G_i^{\text{fake}},
\end{aligned}
$$

$$
\begin{aligned}
\tilde{H}_{k+1} &= \tilde{X}W_1 + B_1 - \gamma \left[I + \tilde{X}\tilde{X}^\top\right] \sum_{i=1}^{k} G_i^{\text{fake}} = \tilde{X}W_1 + B_1 - \gamma \left[I + XX^\top\right] \sum_{i=1}^{k} G_i^{\text{fake}} \\
&= \tilde{H}_1 - \gamma \left[I + XX^\top\right] \sum_{i=1}^{k} G_i^{\text{fake}}.
\end{aligned}
$$

As in Lemma 2, only the first summand $\tilde{X}W_1$ has changed.

And the server cannot distinguish between two different pairs $\{\mathcal{X}, U\}$ if they produce values $\tilde{X}\tilde{X}^\top$.

Therefore, we can replay the conclusion of Lemma 2. $\qquad\square$

### A.3   CUT LAYER

**The proof of Cut Layer Lemma 3.**

*Proof.* Given the client neural network $f$ we denote the last activations before the linear Cut Layer as $Z$. Then, we provide a client with output activations $H = ZW$ and show that the model training protocol remains the same after transforming $Z \to \tilde{Z} = ZU$ and $W_1 \to \tilde{W}_1 = U^\top W_1$.

We define the client's "previous" parameters (before $W$) as $\theta$ and function of this parameters as $f_\theta : f_\theta(\theta, X) = Z$.

Then

$$f(X, \theta, W) = H = f_\theta(\theta, X)W = ZW, \ \mathcal{L} = \mathcal{L}(H) = \mathcal{L}(ZW).$$

Let us consider the gradient of loss $\mathcal{L}$ w.r.t. $W$ and $\theta$:

$$\frac{\partial \mathcal{L}}{\partial W} = \frac{\partial \mathcal{L}}{\partial H}\frac{\partial H}{\partial W} = Z^\top \frac{\partial \mathcal{L}}{\partial H},$$

$$\frac{\partial \mathcal{L}}{\partial \theta} = \frac{\partial \mathcal{L}}{\partial H}\frac{\partial H}{\partial Z}\frac{\partial Z}{\partial \theta} = \left[\frac{\partial Z}{\partial \theta}\right]^* \frac{\partial \mathcal{L}}{\partial H}W^\top = J^* \frac{\partial \mathcal{L}}{\partial H}W^\top,$$

where $J = \frac{\partial Z}{\partial \theta}$ — Jacobian of $f_\theta$.

Thus, after the first two iterations we conclude:

$$H_1 = Z_1 W_1, \quad \theta_2 = \theta_1 - \gamma J_1^* \frac{\partial \mathcal{L}}{\partial H_1}W_1^\top, \quad W_2 = W_1 - \gamma Z_1^\top \frac{\partial \mathcal{L}}{\partial H_1},$$

and

$$H_2 = Z_2 W_2 = f_\theta(\theta_2, X)W_2 = f_\theta(\theta_2, X)W_1 - \gamma f_\theta(\theta_2, X)Z_1^\top \frac{\partial \mathcal{L}}{\partial H_1}.$$

Adding the additional orthogonal matrix $U$ results in:

$$\tilde{W}_1 = U^\top W_1, \quad \tilde{H}_1 = \tilde{Z}_1 \tilde{W}_1 = (Z_1 U)\tilde{W}_1 = H_1, \quad \tilde{W}_2 = \tilde{W}_1 - \gamma \tilde{Z}_1^\top \frac{\partial \mathcal{L}}{\partial H_1}$$

In fact derivative w.r.t $\theta$ is obviously the same, because "trick" was constructed in such way that for any input $Z$ both initial loss and loss after "trick" have the same value $\mathcal{L}(Z) = \tilde{\mathcal{L}}(Z)$, i.e. it is the same function and has the same derivative w.r.t $Z$ and therefore w.r.t $\theta$.

$$\frac{\partial \tilde{\mathcal{L}}}{\partial \theta_1} = \frac{\partial \tilde{\mathcal{L}}}{\partial H_1}\frac{\partial H_1}{\partial \tilde{Z}_1}\frac{\partial \tilde{Z}_1}{\partial Z_1}\frac{\partial Z_1}{\partial \theta} = J_1^* \frac{\partial \mathcal{L}}{\partial H_1}\tilde{W}_1^\top U^\top = J_1^* \frac{\partial \mathcal{L}}{\partial H_1}(U^\top W_1)^\top U^\top$$

$$= J_1^* \frac{\partial \mathcal{L}}{\partial H_1}W_1^\top = \frac{\partial \mathcal{L}}{\partial \theta_1}.$$

Then, for the activations obtained with and without $U$ we claim:

$$\tilde{H}_2 = \tilde{Z}_2 \tilde{W}_2 = f(\theta_2, X)U\tilde{W}_2$$

$$= f(\theta_2, X)U\tilde{W}_1 - \gamma f(\theta_2, X)U\tilde{Z}_1^\top \frac{\partial \mathcal{L}}{\partial H_1}$$

$$= H_2.$$

And that completes the proof.

$\square$

## A.4 CONVERGENCE

**Additional notation.** For matrices, $\|.\|$ is a spectral norm on the space $\mathbb{R}^{n \times m}$, while the $\|.\|_F$ refers to the Frobenius norm.

### A.4.1 "ROTATION" EXAMPLE.

Before we turn to the consideration of the PL-case, we present a simple example for the general non-convex (even 2D) case. We deal with the function $f(y) = y^2 + 6\sin^2 y$, where $y = W^\top X$. We show that convergence of $f$ to the global minimum ($y^* = 0$, $f(y^*) = 0$) breaks after the rotation of "weights" $W$ and "data" $X$, i.e., Adam converges to a local minimum of $f$.

**Example 1.** *Indeed, let the initial weight and data vectors equal:*

$$W = \left(1.915 + \sqrt{2} \cdot 0.6,\ 0\right)^\top,$$

$$X = (1,\ 0)^\top.$$

*We rotate these arguments by an angle of $\frac{\pi}{4}$ with:*

$$U = \frac{1}{\sqrt{2}} \begin{bmatrix} 1 & -1 \\ 1 & 1 \end{bmatrix}.$$

*In addition we pick the learning rate $\gamma = 0.6$. After that, the optimization algorithm stack in the local minima if starting from $(UW,\ UX)$ point. Which we show in Figure 8.*

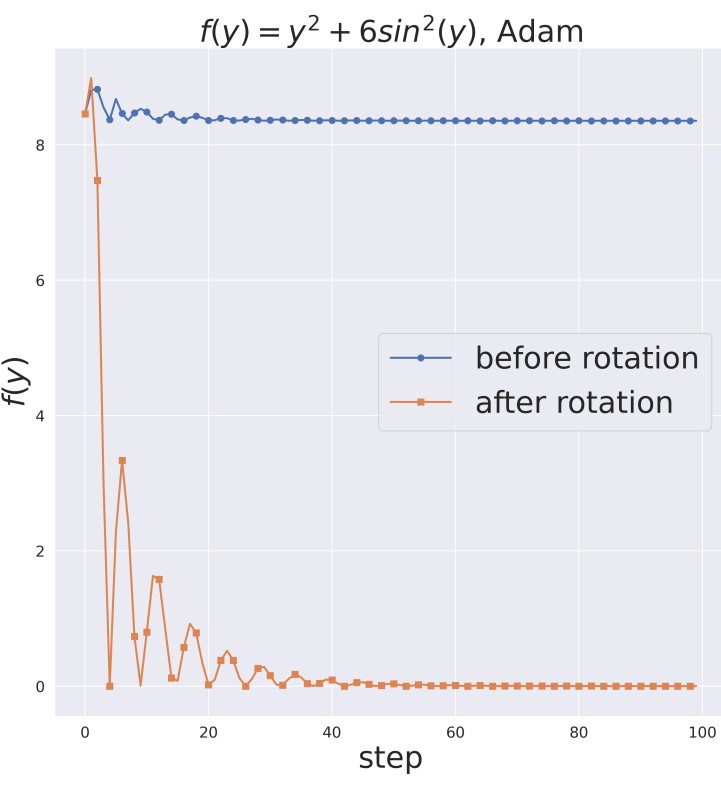

Figure 8: **While optimizing the non-convex function $f(y)$, Adam can get stuck in the local minima depending on the initialization.**

### A.4.2 CONVERGENCE FOR PL FUNCTION. ADAM.

Next, we move to the PL case. In this part of the work we study the convergence of Adam on PL functions under the (semi)orthogonal transformation of data and weights $\{X, W\} \rightarrow \{\tilde{X}, \tilde{W}\} = \{XU, U^\top W\}$.

We need the following assumptions in our discourse.

**Assumption 1** (*L*-smoothness)**.** *Function $\mathcal{L}$ is assumed to be twice differentiable and L-smooth, i.e., $\forall W, W' \in \operatorname{dom} \mathcal{L}$ we have*

$$\|\nabla \mathcal{L}(W) - \nabla \mathcal{L}(W')\| \le L\|W - W'\|.$$

**Assumption 2** (Polyak-Łojasiewicz-condition)**.** *Function $\mathcal{L}$ satisfies the PL-condition if there exists $\mu > 0$, such that for all $W \in \operatorname{dom} \mathcal{L}$ holds*

$$\|\nabla \mathcal{L}(W)\| \ge 2\mu(\mathcal{L}(W) - \mathcal{L}^*),$$

*where $\mathcal{L}^* = \mathcal{L}(W^*)$ is optimal value.*

In addition, we need the assumption on boundedness of gradient (a similar assumptions are made if (Défossez et al., 2022)).

**Assumption 3.** *We assume boundedness of gradient, i.e., for all $W$*

$$\|\nabla \mathcal{L}(W)\| \le \Gamma.$$

In `Adam`(Kingma & Ba, 2017), the bias-corrected first and second moment estimates, i.e. $m_k$ and $\hat{D}_k$ are:

$$m_k = \frac{1 - \beta_1}{1 - \beta_1^k} \sum_{i=1}^{k} \beta_1^{k-i} \nabla \mathcal{L}(W_i), \qquad \hat{D}_k^2 = \frac{1 - \beta_2}{1 - \beta_2^k} \sum_{i=1}^{k} \beta_2^{k-i} \operatorname{diag}\left(\nabla \mathcal{L}(W_i) \odot \nabla \mathcal{L}(W_i)\right).$$

And the starting $m_0$, $\hat{D}_0^2$ (with a little abuse of notation $W_0 = W_1$ in this part only) are given by

$$m_0 = \nabla \mathcal{L}(W_0), \qquad\qquad \hat{D}_0^2 = \operatorname{diag}\left(\nabla \mathcal{L}(W_0) \odot \nabla \mathcal{L}(W_0)\right),$$

with the following update rule:

$$\begin{cases} \hat{m}_{k+1} = \beta_1 \hat{m}_k + (1 - \beta_1)\nabla \mathcal{L}(W_k), \\[2mm] \hat{D}_{k+1}^2 = \beta_2 \hat{D}_k^2 + (1 - \beta_2)\operatorname{diag}(\nabla \mathcal{L}(W_k) \odot \nabla \mathcal{L}(W_k)). \end{cases}$$

Then, the iteration of `Adam` is simply: $W_{k+1} = W_k - \gamma \hat{D}_k^{-1} m_k$.

Despite the promising fact for `(S)GD` that continually many $\{\tilde{X}, \tilde{W}\}$ pairs consistently produce the same activations $\tilde{H} = H$ at each step, we cannot make the same statement for `Adam`-like methods.

**Remark 4.** *Let $\{\tilde{X}, \tilde{W}\} = \{XU, U^\top W\}$ pairs are an orthogonal(semi-orthogonal) transformations of data and weights. Then, these pairs, in general, do not produce the same activations at each step of the Split Learning process with* `Adam`.

*Proof.* Indeed, assume that after k-th step of training, the client substitutes $\{X, W_k\}$ pair with $\{XU, U^\top W_k\}$. Compare the discrepancy between the activations $H_{k+1} = X_{k+1} W_{k+1}$ and $\tilde{H}_{k+1} = \tilde{X}_{k+1}\tilde{W}_{k+1}$. Since, with `Adam`, $\tilde{W}_{k+1} = \tilde{W}_k - \gamma \hat{\tilde{D}}_k^{-1}\hat{\tilde{m}}_k$, the difference arises in the updates of first and second moment (biased) estimates: $\hat{\tilde{D}}_k^{-1}$ and $\hat{\tilde{m}}_k$.

Recall

$$\hat{\tilde{D}}_{\mathrm{k}}^2 - \beta_2 \hat{\tilde{D}}_{\mathrm{k}-1}^2 = (1 - \beta_2) \operatorname{diag}\left( \frac{\partial \mathcal{L}}{\partial \tilde{W}_{\mathrm{k}}} \odot \frac{\partial \mathcal{L}}{\partial \tilde{W}_{\mathrm{k}}} \right)$$

$$= (1 - \beta_2) \operatorname{diag}\left( \frac{\partial \mathcal{L}}{\partial \tilde{H}_{\mathrm{k}}} \frac{\partial \tilde{H}_{\mathrm{k}}}{\partial \tilde{W}_{\mathrm{k}}} \odot \frac{\partial \mathcal{L}}{\partial \tilde{H}_{\mathrm{k}}} \frac{\partial \tilde{H}_{\mathrm{k}}}{\partial \tilde{W}_{\mathrm{k}}} \right)$$

$$= (1 - \beta_2) \operatorname{diag}\left( \tilde{X}^\top \frac{\partial \mathcal{L}}{\partial \tilde{H}_{\mathrm{k}}} \odot \tilde{X}^\top \frac{\partial \mathcal{L}}{\partial \tilde{H}_{\mathrm{k}}} \right)$$

$$\overset{(\tilde{H}_{\mathrm{k}}=H_{\mathrm{k}})}{=} (1 - \beta_2) \operatorname{diag}\left( \tilde{X}^\top \frac{\partial \mathcal{L}}{\partial H_{\mathrm{k}}} \odot \tilde{X}^\top \frac{\partial \mathcal{L}}{\partial H_{\mathrm{k}}} \right)$$

$$= (1 - \beta_2) \operatorname{diag}\left( U^\top X^\top \frac{\partial \mathcal{L}}{\partial H_{\mathrm{k}}} \odot U^\top X^\top \frac{\partial \mathcal{L}}{\partial H_{\mathrm{k}}} \right),$$

the similar holds for $\hat{\tilde{m}}_{\mathrm{k}}$

$$\hat{\tilde{m}}_{\mathrm{k}} - \beta_1 \hat{\tilde{m}}_{\mathrm{k}-1} = (1 - \beta_1) \frac{\partial \mathcal{L}}{\partial \tilde{W}_{\mathrm{k}}} = (1 - \beta_1) \frac{\partial \mathcal{L}}{\partial \tilde{H}_{\mathrm{k}}} \frac{\partial \tilde{H}_{\mathrm{k}}}{\partial \tilde{W}_{\mathrm{k}}} = (1 - \beta_1) \tilde{X}^\top \frac{\partial \mathcal{L}}{\partial \tilde{H}_{\mathrm{k}}}$$

$$\overset{(\tilde{H}_{\mathrm{k}}=H_{\mathrm{k}})}{=} (1 - \beta_1) \tilde{X}^\top \frac{\partial \mathcal{L}}{\partial H_{\mathrm{k}}} = (1 - \beta_1) U^\top X^\top \frac{\partial \mathcal{L}}{\partial H_{\mathrm{k}}}.$$

Then, it is clear how to compare the activations at $\mathrm{k}+1$-th step

$$\begin{cases} \tilde{H}_{\mathrm{k}+1} = \tilde{X} \tilde{W}_{\mathrm{k}+1} = X W_{\mathrm{k}} - \gamma X U \hat{\tilde{D}}_{\mathrm{k}}^{-1} \hat{\tilde{m}}_{\mathrm{k}}, \\[2mm] H_{\mathrm{k}+1} = X W_{\mathrm{k}+1} = X W_{\mathrm{k}} - \gamma X \hat{D}_{\mathrm{k}}^{-1} \hat{m}_{\mathrm{k}}. \end{cases}$$

The difference occurs in $\hat{\tilde{m}}_{\mathrm{k}}$ and $\hat{\tilde{D}}_{\mathrm{k}}^2$. Let us specify

$$\hat{\tilde{D}}_{\mathrm{k}}^2 - \hat{D}_{\mathrm{k}}^2 = (1 - \beta_2)\left[ \operatorname{diag}\left( U^\top X^\top \frac{\partial \mathcal{L}}{\partial H_{\mathrm{k}}} \odot U^\top X^\top \frac{\partial \mathcal{L}}{\partial H_{\mathrm{k}}} \right) - \operatorname{diag}\left( X^\top \frac{\partial \mathcal{L}}{\partial H_{\mathrm{k}}} \odot X^\top \frac{\partial \mathcal{L}}{\partial H_{\mathrm{k}}} \right) \right],$$

for the squared preconditioner, and

$$\hat{\tilde{m}}_{\mathrm{k}} - \hat{m}_{\mathrm{k}} = (1 - \beta_1)\left[ U^\top X^\top \frac{\partial \mathcal{L}}{\partial H_{\mathrm{k}}} - X^\top \frac{\partial \mathcal{L}}{\partial H_{\mathrm{k}}} \right] = (1 - \beta_1)\left[ U^\top - I \right] X^\top \frac{\partial \mathcal{L}}{\partial H_{\mathrm{k}}},$$

for the first moment estimator.

Therefore, a descrepancy between $\tilde{H}_{\mathrm{k}+1}$ and $H_{\mathrm{k}+1}$ vanishes when

$$U \hat{\tilde{D}}_{\mathrm{k}}^{-1} \hat{\tilde{m}}_{\mathrm{k}} = \hat{D}_{\mathrm{k}}^{-1} \hat{m}_{\mathrm{k}}.$$

$\square$

The other thing we can do is analyze whether `Adam`, with the substitution of pairs $\{X, W\} \to \{\tilde{X}, \tilde{W}\}$, can converge to the same optimal value.

Let $W_1$ be an initial weight matrix of the one-layer client linear model from Lemma 1. Our goal is to prove that `Adam` with modifications $\tilde{X} = XU$, $\tilde{W}_1 = U^\top W_1$, where $U$ is proper orthogonal matrix, converges to the same optimum as the original one. Considerations are made similar to (Sadiev et al., 2024).

**Remark 5.** *The difference occurs in the function to be optimized. When we deal with the data and weight transformation we consider the entire model (client-side with the server's top model and loss) as a function of data $X$ and weights $W$. As a function of two variables $\mathcal{L}(X, W) = \mathcal{L}(\tilde{X}, \tilde{W})$, because the activations after the first dense layer of client's model are equal: $\tilde{X}\tilde{W} = XW$. However, in the proof of convergence we are interested only in the dependence of $\mathcal{L}$ on weights $W$. To keep in mind that different functions are optimized, we denote:*

$$\mathcal{L}(W) = \mathcal{L}_{\mathrm{X}}(W),$$

*as a function with input data $X$, and*

$$g(W) = \mathcal{L}_{\tilde{\mathrm{X}}}(W),$$

*as a function with input data $\tilde{X}$. For this case, we require a more stronger assumption than 3:* **Assumption 4** (Modified boundedness of gradient).

$$\forall X, W \quad \|\nabla \mathcal{L}_{\mathrm{X}}(W)\| \leq \Gamma.$$

*Similarly, Assumption 1 and Assumption 2 are required $\forall X$ quantifier, as we assume that $\mathcal{L}$ depends on the data.*

We start with the next technical lemma:

**Lemma 4.** *For all $\mathrm{k} \geq 1$, we have $\alpha I \preccurlyeq \hat{D}_{\mathrm{k}} \preccurlyeq \Gamma I$, where $0 < \alpha \leq \Gamma$.*

*Proof.* Firstly, note that $\hat{D}_{\mathrm{k}}$ is diagonal matrix, where all elements are at least $\alpha$. From this, the property $\alpha I \preccurlyeq \hat{D}_{\mathrm{k}}$ directly follows. Next, we prove the upper bound on $\hat{D}_{\mathrm{k}}$ by induction:

1. **Base case, $\mathrm{k} = 0$:** Using the structure of $\mathrm{diag}\left(\nabla g(\tilde{W}_1) \odot \nabla g(\tilde{W}_1)\right)$, we obtain:

$$
\begin{aligned}
\|\hat{D}_0^2\|_{\infty} &= \left\|\mathrm{diag}\left(\nabla g(\tilde{W}_1) \odot \nabla g(\tilde{W}_1)\right)\right\|_{\infty} \\
&= \left\|\mathrm{diag}\left(U\nabla g(U^{\top}W_1) \odot U\nabla g(U^{\top}W_1)\right)\right\|_{\infty} \\
&= \left\|U\nabla g(U^{\top}W_1)\right\|_{\infty}^2 \leq \left\|U\nabla g(U^{\top}W_1)\right\|_2^2 \\
&= \left\|\nabla g(U^{\top}W_1)\right\|_2^2 \leq \Gamma^2.
\end{aligned}
$$

2. **Induction step, $\mathrm{k} > 0$:** Let $\hat{D}_{\mathrm{k}-1} \preccurlyeq \Gamma I$ by induction hypothesis. Then,

$$
\begin{aligned}
\|\hat{D}_{\mathrm{k}}^2\|_{\infty} &= \left\|\beta_2 \hat{D}_{\mathrm{k}-1}^2 + (1 - \beta_2)\,\mathrm{diag}\left(\nabla g(W_{\mathrm{k}-1}) \odot \nabla g(W_{\mathrm{k}-1})\right)\right\|_{\infty} \\
&\leq \left\|\beta_2 \hat{D}_{\mathrm{k}-1}^2 + (1 - \beta_2)\,\mathrm{diag}\left(\nabla g(W_{\mathrm{k}-1}) \odot \nabla g(W_{\mathrm{k}-1})\right)\right\|_2 \\
&\leq \left\|\beta_2 \hat{D}_{\mathrm{k}-1}^2\right\|_2 + \|(1 - \beta_2)\,\mathrm{diag}\left(\nabla g(W_{\mathrm{k}-1}) \odot \nabla g(W_{\mathrm{k}-1})\right)\|_2 \\
&\leq \left\|\beta_2 \hat{D}_{\mathrm{k}-1}^2\right\|_2 + \|\mathrm{diag}\left(\nabla g(W_{\mathrm{k}-1}) \odot \nabla g(W_{\mathrm{k}-1})\right)\|_2 \\
&\quad - \beta_2 \left\|\mathrm{diag}\left(\nabla g(W_{\mathrm{k}-1}) \odot \nabla g(W_{\mathrm{k}-1})\right)\right\|_2 \\
&\leq \beta_2 \Gamma^2 + \Gamma^2 - \beta_2 \Gamma^2 = \Gamma^2.
\end{aligned}
$$

The same is true for any other $\mathcal{L}_{\mathrm{X}}$. $\qquad\qquad\square$

In the next Lemmas 5 and 6 we omit the dependence on dataset $X$, since that analysis relies only on theresult from Lemma 4 and (modified) assumptions.

**Lemma 5** (Descent Lemma). *Suppose the Assumption 1 holds for function $\mathcal{L}$. Then we have for all $k \geq 0$ and $\gamma$, it is true for* `Adam` *that*

$$\mathcal{L}(W_{k+1}) \leq \mathcal{L}(W_k) + \frac{\gamma}{2\alpha}\|\nabla\mathcal{L}(W_k) - m_k\|^2 - \left(\frac{1}{2\gamma} - \frac{L}{2\alpha}\right)\|W_{k+1} - W_k\|_{\hat{D}_k}^2 - \frac{\gamma}{2}\|\nabla\mathcal{L}(W_k)\|_{\hat{D}_k^{-1}}^2.$$

*Proof.* To begin with, we use $L$-smoothness of the function $\mathcal{L}$ and $\alpha I \preccurlyeq \hat{D}_k$

$$\mathcal{L}(W_{k+1}) \leq \mathcal{L}(W_k) + \langle\nabla\mathcal{L}(W_k), W_{k+1} - W_k\rangle + \frac{L}{2}\|W_{k+1} - W_k\|^2$$

$$\leq \mathcal{L}(W_k) + \langle\nabla\mathcal{L}(W_k), W_{k+1} - W_k\rangle + \frac{L}{2\alpha}\|W_{k+1} - W_k\|_{\hat{D}_k}^2.$$

Applying update $W_{k+1} = W_k - \gamma\hat{D}_k^{-1}m_k$, we obtain

$$\mathcal{L}(W_{k+1}) \leq \mathcal{L}(W_k) + \langle\nabla\mathcal{L}(W_k) - m_k, -\gamma\hat{D}_k^{-1}m_k\rangle + \frac{1}{\gamma}\langle\hat{D}_k(W_k - W_{k+1}), W_{k+1} - W_k\rangle$$

$$+ \frac{L}{2\alpha}\|W_{k+1} - W_k\|_{\hat{D}_k}^2$$

$$= \mathcal{L}(W_k) - \gamma\langle\nabla\mathcal{L}(W_k) - m_k, m_k\rangle_{\hat{D}_k^{-1}} + \frac{1}{\gamma}\langle\hat{D}_k(W_k - W_{k+1}), W_{k+1} - W_k\rangle$$

$$+ \frac{L}{2\alpha}\|W_{k+1} - W_k\|_{\hat{D}_k}^2$$

$$= \mathcal{L}(W_k) + \gamma\langle\nabla\mathcal{L}(W_k) - m_k, \nabla\mathcal{L}(W_k) - m_k\rangle_{\hat{D}_k^{-1}}$$

$$- \gamma\langle\nabla\mathcal{L}(W_k) - m_k, \nabla\mathcal{L}(W_k)\rangle_{\hat{D}_k^{-1}} - \left(\frac{1}{\gamma} - \frac{L}{2\alpha}\right)\|W_{k+1} - W_k\|_{\hat{D}_k}^2$$

$$= \mathcal{L}(W_k) + \gamma\|\nabla\mathcal{L}(W_k) - m_k\|_{\hat{D}_k^{-1}}^2 - \gamma\langle\nabla\mathcal{L}(W_k) - m_k, \nabla\mathcal{L}(W_k)\rangle_{\hat{D}_k^{-1}}$$

$$- \left(\frac{1}{\gamma} - \frac{L}{2\alpha}\right)\|W_{k+1} - W_k\|_{\hat{D}_k}^2.$$

Using denotation $\bar{W}_{k+1} = W_k - \gamma\hat{D}_k^{-1}\nabla\mathcal{L}(W_k)$ and once again update $W_{k+1} = W_k - \gamma\hat{D}_k^{-1}m_k$, we have

$$\mathcal{L}(W_{t+1}) \leq \mathcal{L}(W_k) + \gamma\|\nabla\mathcal{L}(W_k) - m_k\|_{\hat{D}_k^{-1}}^2 - \frac{1}{\gamma}\langle W_{k+1} - \bar{W}_{k+1}, W_k - \bar{W}_{k+1}\rangle_{\hat{D}_k}$$

$$- \left(\frac{1}{\gamma} - \frac{L}{2\alpha}\right)\|W_{k+1} - W_k\|_{\hat{D}_k}^2$$

$$= \mathcal{L}(W_k) + \gamma\|\nabla\mathcal{L}(W_k) - m_k\|_{\hat{D}_k^{-1}}^2 - \left(\frac{1}{\gamma} - \frac{L}{2\alpha}\right)\|W_{k+1} - W_k\|_{\hat{D}_k}^2$$

$$- \frac{1}{2\gamma}\left(\|W_{k+1} - \bar{W}_{k+1}\|_{\hat{D}_k}^2 + \|W_k - \bar{W}_{k+1}\|_{\hat{D}_k}^2 - \|W_{k+1} - W_k\|_{\hat{D}_k}^2\right)$$

$$= \mathcal{L}(W_k) + \gamma\|\nabla\mathcal{L}(W_k) - m_k\|_{\hat{D}_k^{-1}}^2 - \left(\frac{1}{\gamma} - \frac{L}{2\alpha}\right)\|W_{k+1} - W_k\|_{\hat{D}_k}^2$$

$$- \frac{1}{2\gamma}\left(\gamma^2\|\nabla\mathcal{L}(W_k) - m_k\|_{\hat{D}_k^{-1}}^2 + \gamma^2\|\nabla\mathcal{L}(W_k)\|_{\hat{D}_k^{-1}}^2 - \|W_{k+1} - W_k\|_{\hat{D}_k}^2\right).$$

$\square$

Finally, the next lemma concludes our discourse.

**Lemma 6.** *Suppose that the Assumptions 1, 2 for function $\mathcal{L}$ hold. Then we have for all $k \geq 0$ and $\gamma$ the following is true:*

$$\mathcal{L}(W_{k+1}) - \mathcal{L}^* \leq \left(1 - \frac{\gamma\mu}{\Gamma}\right)(\mathcal{L}(W_k) - \mathcal{L}^*)$$
$$+ \frac{\gamma}{2\alpha}\|\nabla\mathcal{L}(W_k) - m_k\|^2$$
$$- \left(\frac{1}{2\gamma} - \frac{L}{2\alpha}\right)\|W_{k+1} - W_k\|^2_{\hat{D}_k}.$$

*Proof.* According to Lemma 5,

$$\mathcal{L}(W_{k+1}) \leq \mathcal{L}(W_k) + \frac{\gamma}{2\alpha}\|\nabla\mathcal{L}(W_k) - m_k\|^2 - \left(\frac{1}{2\gamma} - \frac{L}{2\alpha}\right)\|W_{k+1} - W_k\|^2_{\hat{D}_k} - \frac{\gamma}{2}\|\nabla\mathcal{L}(W_k)\|^2_{\hat{D}_k^{-1}}.$$

With PL-condition and $\frac{1}{\Gamma}I \preccurlyeq \hat{D}_k^{-1}$, we have

$$\mathcal{L}(W_{k+1}) - \mathcal{L}^* \leq \mathcal{L}(W_k) - \mathcal{L}^* + \frac{\gamma}{2\alpha}\|\nabla\mathcal{L}(W_k) - m_k\|^2 - \left(\frac{1}{2\gamma} - \frac{L}{2\alpha}\right)\|W_{k+1} - W_k\|^2_{\hat{D}_k}$$
$$- \frac{\gamma}{2\Gamma}\|\nabla\mathcal{L}(W_k)\|^2$$
$$\leq \mathcal{L}(W_k) - \mathcal{L}^* + \frac{\gamma}{2\alpha}\|\nabla\mathcal{L}(W_k) - m_k\|^2 - \left(\frac{1}{2\gamma} - \frac{L}{2\alpha}\right)\|W_{k+1} - W_k\|^2_{\hat{D}_k}$$
$$- \frac{\gamma\mu}{\Gamma}(\mathcal{L}(W_k) - \mathcal{L}^*).$$

$\square$

## A.5 DIFFERENTIALLY PRIVATE DEFENSE

In addition to obfuscation-based defenses, mentioned in § 1, we consider the DP defense against UnSplit Model Inversion (MI) attack. Before further cogitations, we give several definitions.

**Definition 1.** *Two datasets $X$ and $X'$ are defined as neighbouring if they differ at most in one row.*

**Definition 2.** *Mechanism $\mathcal{M}$ is $(\varepsilon, \delta)$-DP (Differential Private) (Dwork et al., 2006) if for every pair of neighbouring datasets $X \simeq X'$ and every possible output set $S$ the following inequality holds:*
$$\mathbb{P}[\mathcal{M}(X) \in S] \leq e^\varepsilon \mathbb{P}[\mathcal{M}(X') \in S] + \delta.$$

If we call two datasets, $X$ and $X' = XU$, "neighbouring" (It should be noted that this is not the same as Definition 1), the standard Differential Privacy technique is not applicable in this case. Indeed, let us consider two "neighbouring" matrices $X$ and $XU$. We define a subspace of $\mathbb{R}^{n \times n}$ as $S \triangleq \{XX^\top + \alpha \mid \alpha \in [0,1]\}$. Then, for the protocol $\mathcal{A}$ *which does not include randomness itself*, we conclude: $\mathbb{P}(\mathcal{A}(X) \in S) = 1$, $\mathbb{P}(\mathcal{A}(XU) \in S) = 0$ which, in turn, corresponds to $\varepsilon = \infty$, which does not correspond to Definition 2. Thus, including a randomness inside the protocol is essential. Therefore, we use mechanism $\mathcal{M}(X) = \mathcal{A}(X) + Z$ that should satisfy a given differential privacy guarantee for each pair of two neighbouring datasets, already in accordance with the Definition 1. We consider $Z$ is random variable.

Let us explain the estimations on amount of noise required to ensure the mechanism

$$\mathcal{M}(X) = \mathcal{A}(X) + Z \tag{5}$$

satisfies a given differential privacy guarantee. We consider $Z$ is zero mean isotopic Gaussian perturbation $Z \sim \mathcal{N}(0, \sigma^2 I)$.

**Definition 3.** *The global $L_2$ sensitivity of a function $\mathcal{A}$ is following:*

$$\Delta_{\mathrm{F}} = \sup_{X \simeq X'} \|\mathcal{A}(X) - \mathcal{A}(X')\|_{\mathrm{F}}.$$

To estimate the global $L_2$ sensitivity, we plug $\mathcal{A}(X) = X$, take $X$ as an average batch (to cope with the fact that there are often lots of zeroes in each batch), take $X'$ as $X$ with replacement of a random row with zeroes, and then perform the calculation of $\Delta_F$ as in Definition 3.

**Theorem 1** (see Theorem 8 from (Balle & Wang, 2018)). *Let $\mathcal{A}$ be a function with global $L_2$ sensitivity $\Delta_F$. For and $\varepsilon > 0$ and $\delta \in (0, 1)$, the mechanism described in Algorithm 1 from (Balle & Wang, 2018) is $(\varepsilon, \delta)$-DP.*

Thus, in our code we add implementation of Algorithm 1 from (Balle & Wang, 2018) to estimate $\sigma$, then apply the mechanism (5) and, finally, do experiments with Differential Privacy.

We perform numerical experiments to test Differential Privacy defence against an attack from (Erdoğan et al., 2022). See results in Table 4. As one may note, in this case defence presents, but is far from perfect.

## B    DISCUSSION OF THE DEFENSE FRAMEWORKS

The development of attacks and defensive strategies progresses in parallel. Corresponding defense strategies are broadly categorized into cryptographic (Cheng et al., 2021; Liu et al., 2022; Bonawitz et al., 2017; Yang et al., 2019a; Hardy et al., 2017; Smith et al., 2020) and non-cryptographic defenses. The former technique provides a strict privacy guarantee, but it suffers from computational overhead (Erdoğan et al., 2022; Zou et al., 2022; Nguyen et al., 2023c) during training which is undesirable in the context of SL. Consequently, cryptographic defenses are not considered in our work. Non-cryptographic defenses include: Differential Privacy (DP) mechanisms (Dwork & Roth, 2014; Hu et al., 2022; Li et al., 2022c; Mi et al., 2023), data obfuscation (Gu et al., 2023; 2020; Sun et al., 2021; Wang et al., 2022; Qiu et al., 2024b) and adversarial training (Li et al., 2022a; Wang et al., 2023; Vepakomma et al., 2020; Sun et al., 2021; Turina et al., 2021; Erdogan et al., 2023; Yang et al., 2019b; Gao & Zhang, 2023). Since the DP solution does not always satisfy the desired privacy-utility trade-off for the downstream task (Ghazi et al., 2021; Zou et al., 2023), the most common choice (Yu et al., 2024) is to use data obfuscation and adversarial training strategies to prevent the feature reconstruction attack.

## C    ADDITIONAL EXPERIMENTS

**MSE across different classes.** In addition to Figures 1 and 3 to 6, we also report the Cut Layer ($\mathcal{Z}$-space) and Reconstruction ($\mathcal{X}$-space) mean square error dependencies on the exact number of classes. Measure MSE w.r.t. each class is much more convenient and computationally faster than the analogues experiment for FID.

We show those MSE between "true" and reconstructed features after the UnSplit attack on CIFAR-10, F-MNIST and MNSIT in Figure 9. As, generally, MSE do not describe the reconstruction quality by itself (unlike the FID, which can be validated with benchmarks (Jayasumana et al., 2024)), we observe that MSE for MLP-based models are, approximately, 10 times largen than for CNN-based, meaning that the discrepancy between the original and reconstructed data should be more expressive in case of client-side model filled with dense layers. At the same time, an interesting thing we see in Figure 9: Cut Layer MSE is much more lower than Reconstruction MSE for the MLP-based model architecture, while these MSE values are approximately the same for CNNs. This fact reflects the Cut Layer Lemma 3.

Provided figures contribute to the knowledge of which classes in the dataset are more "sensitive" to feature reconstruction attacks. We also highlight that combining these findings with any of the defense methods may significantly weaken the attacker's side, or(and) the non-label party can concentrate on the defense of specific classes from the dataset.

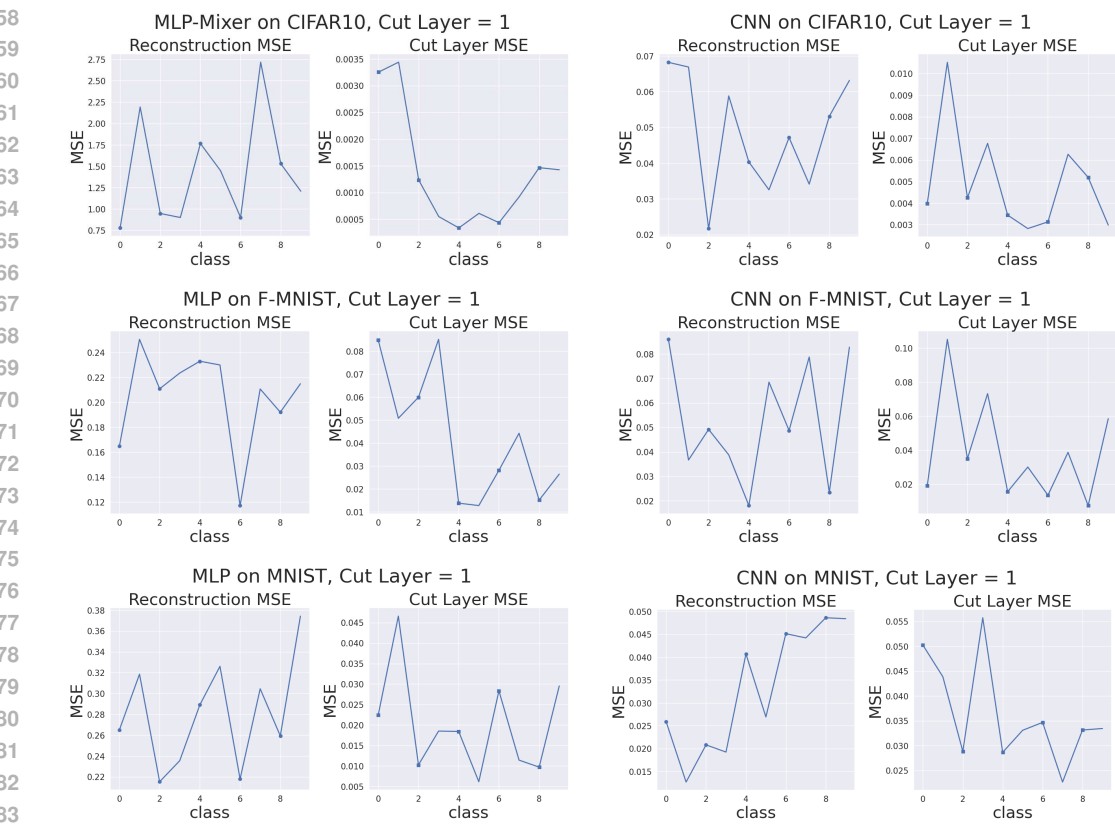

Figure 9: **MSE across different classes for the UnSplit attack.** (**Top row**): CIFAR-10 — MLP-Mixer and CNN-based models. (**Middle row**): F-MNIST — MLP and CNN-based models. (**Bottom row**): MNIST — MLP and CNN-based models.

**Architectural design.** Since we claim that feature reconstruction attacks are not useful against MLP-based models, we aim to alleviate certain concerns about the experimental part of our work. In particular, we answer the following question: "What is the exact architectural design of the MLP-based models?"

The exact architectures are available in our repository. Additionally, we specify that it is a four-layer MLP with `ReLU()` activation functions for MNIST and F-MNIST datasets, and we use the PyTorch (Paszke et al., 2019) MLP-Mixer implementation from [3] repository. All CNN-based models are listed in the original UnSplit and FSHA papers' repositories [4] and [5]. Nevertheless, we should stress that the main findings of our paper *are independent of any specific architecture and work with any MLP-based model*. The key requirement is that the architecture should consist mostly of linear layers and activation functions between them, thus, convolutional layers should be omitted.

**Experiments on small MLP model.** In addition to our main experiments with MLP-based architectures Table 1 and Figures 1, 3 to 6 and 9, we also report results from experiments using a small MLP-based model (SmallMLP). Unlike the models in Table 1, the SmallMLP architecture does not match the accuracy of the CNN-based model but is designed to match its number of parameters.

| # Parameters / Model | MLP | MLP-Mixer | CNN | SmallMLP |
|---|---|---|---|---|
| # | 2,913,290 | 146,816 | 45,278 | 7,850 |

Table 2: **Number of parameters for different models across.**

---

[3] https://github.com/omihub777/MLP-Mixer-CIFAR
[4] https://github.com/ege-erdogan/unsplit
[5] https://github.com/pasquini-dario/SplitNN_FSHA

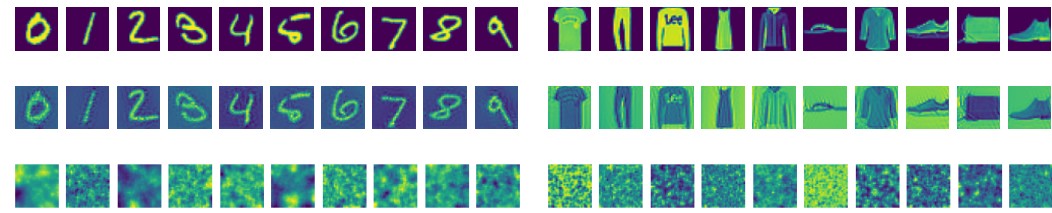

Figure 10: Results of UnSplit attack on MNIST. Figure 11: Results of UnSplit attack on F-MNIST. (**Top**): Original images. (**Middle**): CNN-based (**Top**): Original images. (**Middle**): CNN-based client model. (**Bottom**): **SmallMLP** client model. client model. (**Bottom**): **SmallMLP** client model.

Specifically, the SmallMLP architecture consists of a two-layer linear model with a `ReLU()` activation function. The client's part of the model includes one linear layer, therefore, we follow the conditions of Lemma 1 in this case.

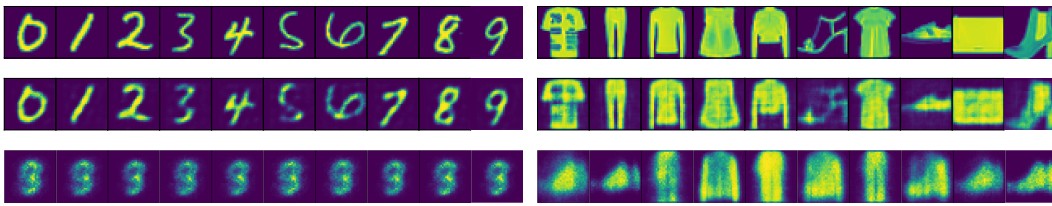

Figure 12: Results of FSHA attack on MNIST. Figure 13: Results of FSHA attack on F-MNIST. (**Top**): Original images. (**Middle**): CNN-based (**Top**): Original images. (**Middle**): CNN-based client model. (**Bottom**): **SmallMLP** client model. client model. (**Bottom**): **SmallMLP** client model.

As we observe in Figures 10 and 11, even a small MLP-based model with less number of parameters than the CNN model remains resistant to the UnSplit attack. The same for the FSHA attack we conclude from Figures 12 and 13. However, the accuracy of SmallMLP is significantly lower compared to our four-layer MLP, with an average accuracy of **92.6% vs. 98.5%** on MNIST, and **83.8% vs. 88.3%** on F-MNIST.

**ViT & ResNet on Tiny ImageNet.** The following description complements our results in Figure 2. We have conducted runs of ResNet-18 He et al. (2015) and ViT Dosovitskiy et al. (2021) when training on Tiny ImageNet Wu et al. (2017) under FSHA. Tiny ImageNet is a smaller version of the ImageNet dataset, designed for efficient evaluation of object classification models. It contains 200 object categories, each with 500 training images, 50 validation images, and 50 test images, all resized to $64 \times 64$ pixels. For ViT, the client side contains the embedding layer and 1 transformer layer with `hidden_size` $= 192$, 6 attention heads, and `intermediate_size` $= 768$. For ResNet, the client side contains the first 2 layers (2 blocks) to approximately match the number of parameters with the ViT case. For

| Model | Acc@1(%) | Acc@5(%) |
|---|---|---|
| ResNet-18 | 6.63 | 15.13 |
| ViT | 1.03 | 4.07 |

Table 3: **Acc. of the pre-trained classifier on images, reconstructed after FSHA, when client-side model is: (I) CNN-based ResNet; (II) ViT.** For Tiny Imagenet the random guessing is 0.5%.

the decoder and discriminator, we experimented with both convolutional-based and attention-based architectures. We conducted the attack for 5000 steps with a batch size of 32 and tested 4 different learning rate values for the optimizers: $\{3e^{-4}, 1e^{-4}, 3e^{-5}, 1e^{-5}\}$ (note that following the original paper, we set the discriminator learning rate 10 times higher than for the other networks). We also report an accuracy@1 and accuracy@5 a pre-trained Tiny ImageNet classifier tested on the reconstructed after the attack (i.e., after 5000 steps of FSHA) images—see Table 3. "Acc@1" and "Acc@5" refer to top-1 and top-5 accuracy metrics, where Acc@1 measures the percentage of samples for which the model's highest confidence prediction matches the true label, and Acc@5 measures the percentage of samples for which the true label appears among the model's top 5 highest confidence predictions. As a classifier, we used an open source ViT model from HuggingFace models [6], which achieves Acc@1: 79.65% and Acc@5: 93.12% on Tiny ImageNet. As evident from the results, the ViT-based client demonstrates significantly better performance compared to ResNet-18 — only

---

[6]tzhao3/vit-tiny-imagenet

slightly above the random guess level, which is $0.5\%$ for 200 classes — once again confirming the theoretical insights obtained in our paper.

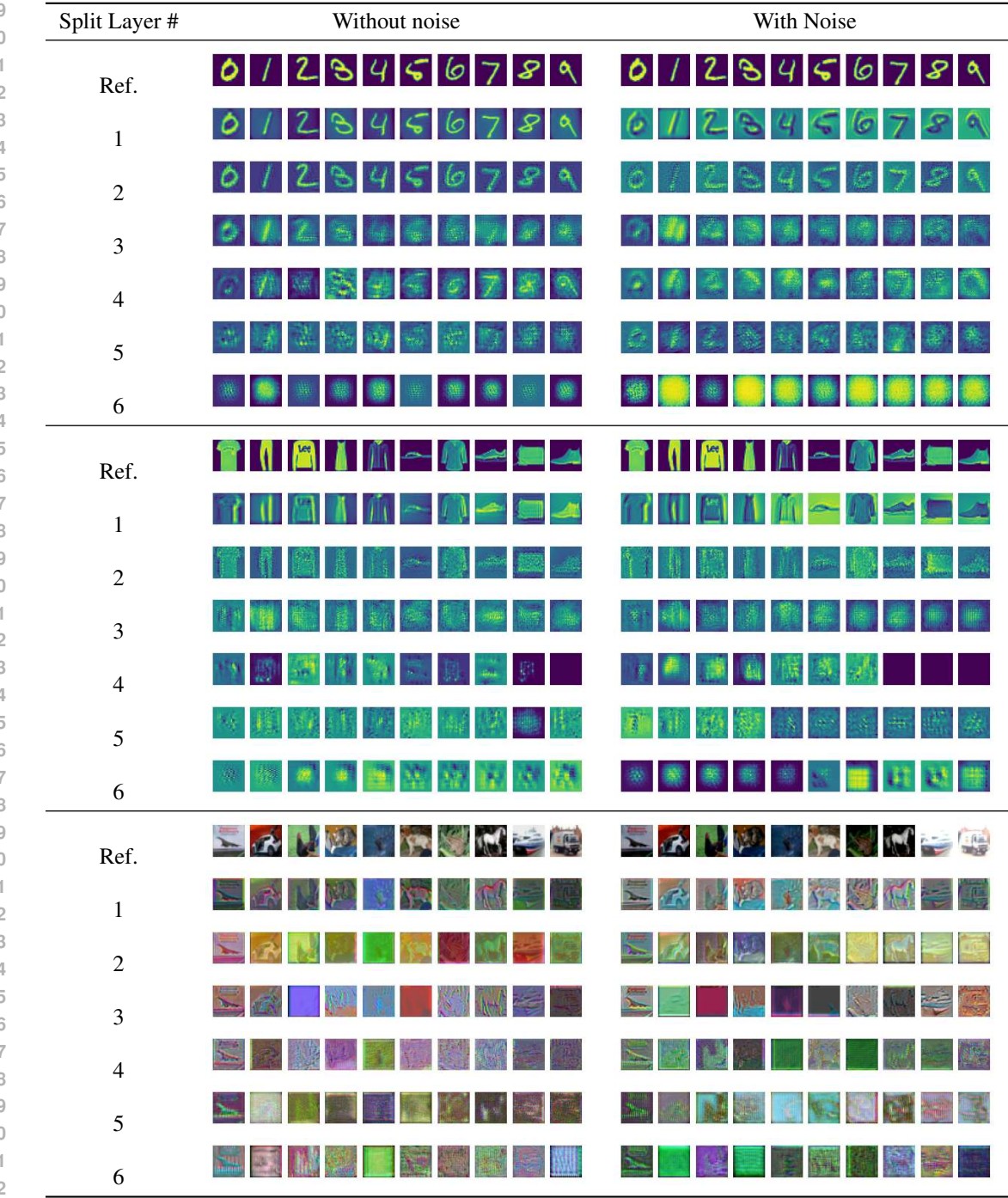

Table 4: **Estimated inputs with and without adding noise for various Cut Layers for the MNIST, F-MNIST, and CIFAR-**10 **datasets.** The "Ref." row display the actual inputs, and the next rows display the attack results for different split depths. We took the following noise variance for different datasets: $\sigma = 1.6$ for MNIST, $\sigma = 2.6$ for F-MNIST, $\sigma = 0.25$ for CIFAR-10. Note that theoretical value of $\sigma$ for CIFAR-10 is 7.1, but we decided to lower it due to neural network learning issues.

