# OpenReview forum: "Just a Simple Transformation is Enough for Data Protection in Split Learning"
_ICLR.cc/2026/Conference — ICLR 2026 Conference Withdrawn Submission_

### Official Review · Reviewer_ic93 · 2025-10-19

**Soundness:** 1
**Presentation:** 1
**Contribution:** 1
**Rating:** 2
**Confidence:** 5

**Summary:**

This paper analyzes the differences in data reconstruction attacks between CNN and MLP models in split learning systems and concludes that MLP models are more resistant to such attacks than CNN models.

**Strengths:**

The study topic is good, but I can no longer identify any notable strengths.

**Weaknesses:**

1.The paper confuses the two scenarios, VFL and Two-Party SL. In previous works [1,2], VFL typically adopts a two-party setting where the client holds part of the model (the bottom model), and the server also holds partial data and a portion of the bottom model. The intermediate representations from both sides are aggregated and passed through the server’s top model. In contrast, Two-Party SL [3,4,5,6,7] is fundamentally different: the server only holds the latter part of the model and has no access to private data—it merely receives intermediate representations and returns the corresponding gradients to the client. The scenario studied in this paper is SL, not VFL, as clearly shown in Section 3 (Setup).

2.The paper does not evaluate state-of-the-art attacks. It only considers UnSplit [3] and FSHA [4], which are outdated methods developed before 2022. More recent and more powerful attacks, such as PCAT [5], FORA [6], and SDAR [7], are not included in the evaluation. These newer attacks have demonstrated superior performance compared to UnSplit and FSHA and may not experience the same poor results on MLP models. Moreover, as shown in [8], attackers can also achieve effective data reconstruction using a ViT architecture.

3.The remarks in Section 3 lack proper justification. In Remarks 1 and 3, the paper directly claims that the attacker “cannot reconstruct the initial data X” but this conclusion is neither theoretically proven nor experimentally verified. Moreover, the attacker is unaffected by the client’s transformations on the model or input—as long as the activations ( H ) remain the same, nothing changes from the attacker’s perspective. Additionally, many attack methods do not rely on matching data distributions; they can still perform effectively even when using auxiliary data with different distributions (as shown in FSHA [4], FORA [6], and SDAR [7]).

4.FID is already a widely used metric in model inversion attacks. While the paper claims that FID is superior to MSE, MSE is not specifically designed for image-level evaluation. What about other commonly used metrics such as SSIM, PSNR, or LPIPS? In Table 1, the MSE of the MLP-based model is better than that of the CNN on F-MNIST, while the FID of the MLP-based model is better than CNN on CIFAR-10. These results are inconsistent with the paper’s conclusions.

5.The cut layer is set too shallow in the experiments, with only layer 1 evaluated, as UnSplit performs poorly on deeper layers. To convincingly demonstrate that attacks on MLP models are less effective than on CNNs, a range of cut layers should be tested for a comprehensive comparison. Additionally, to support the claim that MLP models are harder to attack than CNNs, experiments should include diverse MLP architectures, and the client could also add several MLP layers to existing CNN models for evaluation.

6.Questions arise regarding the results. In Figure 5, the attack results on the MLP-based model do not match the original images; for example, an original “0” or shoe is reconstructed as “1” or a T-shirt. If the attacker were less effective, the output should appear blurry rather than completely mismatched. Similarly, the results in Figure 7 suggest inappropriate hyperparameter settings rather than a fundamental weakness of the attack—the hyperparameters may need adjustment for different models, as the attack does not appear fully optimized.

7.Minor Issues: Many references lack conference or journal names and are therefore incomplete. The paper does not disclose significant LLM usage, as required by the author guidelines.

[1]Fu, Chong, et al. "Label inference attacks against vertical federated learning." 31st USENIX security symposium (USENIX Security 22). 2022.

[2]Yao, Duanyi, et al. "URVFL: Undetectable Data Reconstruction Attack on Vertical Federated Learning.". NDSS 2025.

[3]Erdoğan, Ege, Alptekin Küpçü, and A. Ercüment Çiçek. "Unsplit: Data-oblivious model inversion, model stealing, and label inference attacks against split learning." Proceedings of the 21st Workshop on Privacy in the Electronic Society. 2022.

[4]Pasquini, Dario, Giuseppe Ateniese, and Massimo Bernaschi. "Unleashing the tiger: Inference attacks on split learning." Proceedings of the 2021 ACM SIGSAC conference on computer and communications security. 2021.

[5]Gao, Xinben, and Lan Zhang. "{PCAT}: Functionality and data stealing from split learning by {Pseudo-Client} attack." 32nd USENIX Security Symposium (USENIX Security 23). 2023.

[6]Xu, Xiaoyang, et al. "A stealthy wrongdoer: Feature-oriented reconstruction attack against split learning." Proceedings of the IEEE/CVF conference on computer vision and pattern recognition. 2024.

[7]Zhu, Xiaochen, et al. "Passive inference attacks on split learning via adversarial regularization." NDSS 2025.

[8]Wang, Lixu, et al. "Split adaptation for pre-trained vision transformers." Proceedings of the Computer Vision and Pattern Recognition Conference. 2025.

**Questions:**

Please see the above Weaknesses.

---

### Official Review · Reviewer_L3NX · 2025-10-29

**Soundness:** 1
**Presentation:** 2
**Contribution:** 1
**Rating:** 2
**Confidence:** 5

**Summary:**

The paper found that in split learning, the MLP-mixer is enough to defend against inversion attacks in split learning frameworks as more than 1 layer is placed on the client side.

**Strengths:**

1. The paper empirically shows that for visual tasks such as image classification, leveraging MLP and transformers are sufficient to defend against inversion attacks in split learning

**Weaknesses:**

1. I think authors have a misunderstanding of vertical federated learning and split learning. But it is fine to focus on split learning.
2. I am confused about the motivation. For split learning, a portion of the application is image classification and it is common to use CNNs. It is also verified that a vision transformer will leak privacy [1]. Why do authors focus on MLP? What is the motivation for using MLP for visual tasks? If the paper is going to discuss MLP, related work on transformers should be discussed and compared [2], [4].
3. The three main contribution. 1: Without prior data knowledge, split learning is privacy-preserving enough. 2: MLP based model are sufficient to defense against inversion attacks. 3: using human-feeling central metrics, are already discussed previously in the NLP fields [2]. The difference is that this paper focuses on vision. But again, what is the motivation for using MLP in vision? The reason why contribution 1 stands out in NLP is because it is basically impossible to find similar prior knowledge of data distribution in the real world. But for image, it is common. There is a lot of medical image application and human face identification systems in real-world using split learning. You can easily get public datasets of medical images and human faces. The assumption that attackers have no chance to know prior data distribution does not hold. See examples in [1] and [3].
4. The paper's main contribution: Linear layer and random dropout can preserve privacy is already well-discussed in previous literature [1], [2],
5. Experiments are weak. More practical datasets such as CelebA, Coco should be evaluated.

[1] Xu, Hengyuan, Liyao Xiang, Hangyu Ye, Dixi Yao, Pengzhi Chu, and Baochun Li. "Permutation equivariance of transformers and its applications." In Proceedings of the IEEE/CVF Conference on Computer Vision and Pattern Recognition, pp. 5987-5996. 2024.
[2] Yao, Dixi, and Baochun Li. "Is Split Learning Privacy-Preserving for Fine-Tuning Large Language Models?." IEEE Transactions on Big Data (2024).
[3] Ghosh, Bishwamittra, Yuan Wang, Huazhu Fu, Qingsong Wei, Yong Liu, and Rick Siow Mong Goh. "Split learning of multi-modal medical image classification." In 2024 IEEE Conference on Artificial Intelligence (CAI), pp. 1326-1331. IEEE, 2024.
[4] Oh, Seungeun, Sihun Baek, Jihong Park, Hyelin Nam, Praneeth Vepakomma, Ramesh Raskar, Mehdi Bennis, and Seong-Lyun Kim. "Privacy-preserving split learning with vision transformers using patch-wise random and noisy cutmix." arXiv preprint arXiv:2408.01040 (2024).

**Questions:**

1. Why is the accuracy for MNIST and CIFAR10 so low? ViT should be able to reach 94% on CIFAR10 and 99% on MNIST without any pre-training.
2. Why is FID for CNN higher than that for MLP?
3. What models do you use and at which layers do you cut the model?

**Details Of Ethics Concerns:**

There are no ethics issues regarding privacy and other related problems.

---

### Official Review · Reviewer_4ZxJ · 2025-10-29

**Soundness:** 3
**Presentation:** 3
**Contribution:** 3
**Rating:** 4
**Confidence:** 3

**Summary:**

This paper studies privacy protection in Split Learning (SL), focusing on feature reconstruction attacks such as Model Inversion (MI) and Feature-space Hijacking (FSHA). The authors propose a simple yet powerful idea: Merely changing the client-side model architecture (replacing CNNs with MLP-based models) can inherently improve privacy protection — even without any additional defense mechanisms.
Extensive experiments demonstrate that MLP-based models can effectively defend against state-of-the-art feature reconstruction attacks.

**Strengths:**

The paper is clearly written, logically organized, and supported by carefully reasoned formal proofs that effectively illustrate both the motivation and the method.
The paper introduces an elegant idea — structural transformation itself can serve as privacy protection and shows that Hijacking and Model Inversion attacks fail on MLP-based models
without any additional changes.
The authors point out the limitations of MSE for assessing image privacy and introduce FID as a more perceptually faithful measure.

**Weaknesses:**

The claim of “the findings can be combined with any of the existing defense frameworks” is appealing but lacks experiments comparing with known defenses.
The experiment only focuses on image datasets and lacks text datasets.

**Questions:**

Could you please provide experimental results comparing with or combining with the existing defense frameworks?
Please provide small-scale experiments with text or tabular data to strengthen the generalizability of your argument.

---

### Official Review · Reviewer_Cxar · 2025-11-12

**Soundness:** 2
**Presentation:** 2
**Contribution:** 2
**Rating:** 2
**Confidence:** 4

**Summary:**

This paper is set in the context of malicious attacks in a Split Learning setup.
In particular, the authors discuss the feature reconstruction attacks like Model
Inversion (MI) and Feature Space Hijacking Attack (FSHA). The authors provided
a detailed background study and claimed that these attacks cannot be
successful without prior knowledge of the distribution of the data. They also
show that theoretically, for a single dense layer at the client side, it is possible
to prevent such attacks using a simple translation using an orthonormal matrix.
The authors also argue that models depending on dense layers prevent
reconstruction to a much greater extent compared to convolutional networks.
Overall, the paper is well-written and the arguments are clear and precise. The
authors provided some experiments showcasing the same attacks on different
models to establish their claims.

**Strengths:**

(1) The paper establishes that an orthogonal transformation to the inputs and
weights of the last layer makes it impossible to reconstruct features from
the final activation at the cut layer.
(2) The paper provides a detailed background study and establishes the claims
and arguments precisely. The paper is well-written.
(3) The paper argues and introduces FID as a better metric for the success
of reconstruction attacks.
(4) The paper also establishes that it is impossible for the attacker to reconstruct
the inputs without the knowledge of prior data distribution.

**Weaknesses:**

(1) The paper makes a compelling case for the simple transformation using the
orthogonal matrix. However, it is unclear how the transformation affects
the underlying distribution of the activations and weights. If the distributions
do not change with the transformations, it may still be possible to
recover some information at the attacker’s end.
(2) The authors theoretically establish the efficacy of the transformations for
a single layer. They also argue that in the case of an MLP, the layers
before the last layer would change the distribution of data, making the
feature reconstruction impossible. It would be nice to see this happening
empirically.
(3) The authors claim that the FID is a better metric than MSE for reconstruction.
However, it is not clear why. While FID is often used to measure
perceptual quality and fidelity in generative models, it does not guarantee
any measurement of the information content. And for the task at hand,
information content seems to be more important than fidelity or perceptual
quality. To elaborate, FID may measure pixel-level distribution of
original and reconstructed images. However, it is not guaranteed that
the poorly constructed dissimilar images will pass enough information to
understand the latent distribution.
(4) The authors showed empirical results on CNNs and MLPs using MSE
and FID, and claimed that MLPs are better in the defense of feature
reconstruction attacks. However, from the results in Table 1, it is very
unclear for two of the three datasets authors have showcased. For FMNIST
and CIFAR-10, the MSE and FID metrics, respectively, show
that CNN has a better defence than MLP.
(5) Although the authors claim that the MLPs are better than CNNs in such
defence, there is no discussion of why CNNs fail.
2

**Questions:**

See above

---

### Note · Authors · 2025-11-14

**Comment:**

We thank the Reviewers for time, and their feedback. I have read and agree with the venue's withdrawal policy on behalf of myself and my co-authors.

**Withdrawal Confirmation:**

I have read and agree with the venue's withdrawal policy on behalf of myself and my co-authors.